# Calibrating Predictions to Decisions: A Novel Approach to Multi-Class Calibration

**Shengjia Zhao**
Stanford University
sjzhao@stanford.edu

**Michael P. Kim**
UC Berkeley
mpkim@berkeley.edu

**Roshni Sahoo**
Stanford University
rsahoo@stanford.edu

**Tengyu Ma**
Stanford University
tengyuma@stanford.edu

**Stefano Ermon**
Stanford University
ermon@stanford.edu

## Abstract

When facing uncertainty, decision-makers want predictions they can trust. A machine learning provider can convey confidence to decision-makers by guaranteeing their predictions are distribution calibrated — amongst the inputs that receive a predicted class probabilities vector $q$, the actual distribution over classes is $q$. For multi-class prediction problems, however, achieving distribution calibration tends to be infeasible, requiring sample complexity exponential in the number of classes $C$. In this work, we introduce a new notion—*decision calibration*—that requires the predicted distribution and true distribution to be "indistinguishable" to a set of downstream decision-makers. When all possible decision makers are under consideration, decision calibration is the same as distribution calibration. However, when we only consider decision makers choosing between a bounded number of actions (e.g. polynomial in $C$), our main result shows that decisions calibration becomes feasible — we design a recalibration algorithm that requires sample complexity polynomial in the number of actions and the number of classes. We validate our recalibration algorithm empirically: compared to existing methods, decision calibration improves decision-making on skin lesion and ImageNet classification with modern neural network predictors.

## 1 Introduction

Machine learning predictions are increasingly employed by downstream decision makers who have little or no visibility on how the models were designed and trained. In high-stakes settings, such as healthcare applications, decision makers want predictions they can trust. For example in healthcare, suppose a machine learning service offers a supervised learning model to healthcare providers that claims to predict the probability of various skin diseases, given an image of a lesion. Each healthcare provider want assurance that the model's predictions lead to beneficial decisions, according to their own loss functions. As a result, the healthcare providers may reasonably worry that the model was trained using a loss function different than their own. This mismatch is often inevitable because the ML service may provide the same prediction model to many healthcare providers, which may have different treatment options available and loss functions. Even the same healthcare provider could have different loss functions throughout time, due to changes in treatment availability.

If predicted probabilities perfectly equal the true probability of the event, this issue of trust would not arise because they would lead to optimal decision making regardless of the loss function or task considered by downstream decision makers. In practice, however, predicted probabilities are never perfect. To address this, the healthcare providers may insist that the prediction function be

35th Conference on Neural Information Processing Systems (NeurIPS 2021).

*distribution calibrated*, requiring that amongst the inputs that receive predicted class probability vectors $q$, the actual distribution over classes is $q$. This solves the trust issue because among the patients who receive prediction $q$, the healthcare providers knows that the true label distribution is $q$, and hence knows the true expected loss of a treatment on these patients. Unfortunately, to achieve distribution calibration, we need to reason about the set of individuals $x$ who receive prediction $q$, for *every* possible predicted $q$. As the number of distinct predictions may naturally grow exponentially in the number of classes $C$, the amount of data needed to accurately certify distribution calibration tends to be prohibitive. Due to this statistical barrier, most work on calibrated multi-class predictions focuses on obtaining relaxed variants of calibration. These include *confidence calibration* (Guo et al., 2017), which calibrates predictions only over the most likely class, and *classwise calibration* (Kull et al., 2019), which calibrates predictions for each class marginally. While feasible, these notions are significantly weaker than distribution calibration and do not address the trust issue highlighted above. Is there a calibration notion that addresses the issue of trust, but can also be verified and achieved efficiently? Our paper answers this question affirmatively.

**Our Contributions.** We introduce a new notion of calibration—*decision calibration*—where we take the perspective of potential decision-makers: the only differences in predictions that matter are those that could lead to different decisions. Inspired by Dwork et al. (2021), we formalize this intuition by requiring that predictions are "indistinguishable" from the true outcomes, according to a collection of decision-makers.

First, we show that prior notions of calibration can be characterized as special cases of decision calibration under different collections of decision-makers. This framing explains the strengths and weakness of existing notions of calibration, and clarifies the guarantees they offer to decision makers. For example, we show that a predictor is distribution calibrated if and only if it is decision calibrated with respect to *all* loss functions and decision rules. This characterization demonstrates why distribution calibration is so challenging: achieving distribution calibration requires simultaneously reasoning about all possible decision tasks.

The set of *all* decision rules include those that choose between exponentially (in number of classes $C$) many actions. In practice, however, decision-makers typically choose from a bounded (or slowly-growing as a function of $C$) set of actions. Our main contribution is an algorithm that *guarantees decision calibration for such more realistic decision-makers*. In particular, we give a sample-efficient algorithm that takes a pre-trained predictor and post-processes it to achieve decision calibration with respect to *all* decision-makers choosing from a bounded set of actions. Our recalibration procedure does not harm other common performance metrics, and actually improves accuracy and likelihood of the predictions. In fact, we argue formally that, in the setting of bounded actions, optimizing for decision calibration recovers many of the benefits of distribution calibration, while drastically improving the sample complexity. Empirically, we use our algorithm to recalibrate deep network predictors on two large scale datasets: skin lesion classification (HAM10000) and Imagenet. Our recalibration algorithm improves decision making, and allow for more accurate decision loss estimation compared to existing recalibration methods even under distribution shift.

## 2 Background

### 2.1 Setup and Notation

We consider the prediction problem with random variables $X$ and $Y$, where $X \in \mathcal{X}$ denotes the input features, and $Y \in \mathcal{Y}$ denotes the label. We focus on classification where $\mathcal{Y} = \{(1, 0, \cdots, 0), (0, 1, \cdots, 0), \cdots, (0, 0, \cdots, 1)\}$ where each $y \in \mathcal{Y}$ is a one-hot vector with $C \in \mathbb{N}$ classes. [1] A probability prediction function is a map $\hat{p} : \mathcal{X} \to \Delta^C$ where $\Delta^C$ is the $C$-dimensional simplex. We define the support of $\hat{p}$ as the set of distributions it could predict, i.e. $\{\hat{p}(x)|x \in \mathcal{X}\}$. We use $p^* : \mathcal{X} \to \Delta^C$ to denote the true conditional probability vector, i.e. $p^*(x) = \mathbb{E}[Y \mid X = x] \in \Delta^C$, and for all $c \in [C]$, each coordinate gives the probability of the class $p^*(x)_c = \Pr[Y_c \mid X = x]$.

---

[1] We can also equivalently define $\mathcal{Y} = \{1, 2, \cdots, C\}$, here we denote $y$ by a one-hot vector for notation convenience when taking expectations.

## 2.2 Decision-Making Tasks and Loss Functions

We formalize a decision-making task as a loss minimization problem. The decision-maker has some set of available actions $\mathcal{A}$ and a loss function $\ell : \mathcal{Y} \times \mathcal{A} \to \mathbb{R}$. In this paper we assume the loss function does not directly depend on the input features $X$. For notational simplicity we often refer to a (action set, loss function) pair $(\mathcal{A}, \ell)$ only by the loss function $\ell$: the set of actions $\mathcal{A}$ is implicitly defined by the domain of $\ell$. We denote the set of all possible loss functions as $\mathcal{L}_{\text{all}} = \{\ell | \forall \text{ set } \mathcal{A} \text{ and function } \ell : \mathcal{Y} \times \mathcal{A} \to \mathbb{R}\}$.

We treat all action sets $\mathcal{A}$ with the same cardinality as the same set — they are equivalent up to renaming the actions. A convenient way to think about this is that we only consider actions sets $\mathcal{A} \in \{[1], [2], \cdots, [K], \cdots, \mathbb{N}, \mathbb{R}\}$ where $[K] = \{1, \cdots, K\}$.

## 2.3 Bayes Decision-Making

Given some predicted probability $\hat{p}(X)$ on $Y$, a decision-maker selects an action in $\mathcal{A}$. We assume that the decision-maker selects the action based on the predicted probability. That is, we define a decision function as any map from the predicted probability to an action $\delta : \Delta^C \to \mathcal{A}$. and denote by $\mathbb{A}_{\text{all}}$ as the set of all decision functions with any set of actions $\mathbb{A}_{\text{all}} = \{\delta \mid \forall \text{ set } \mathcal{A} \text{ and function } \delta : \Delta^C \to \mathcal{A}\}$.

Typically a decision-maker selects the action that minimizes the expected loss (under the predicted probability). This strategy is formalized by the following definition of Bayes decision-making.

**Definition 1** (Bayes Decision). *Choose any $\ell \in \mathcal{L}_{\text{all}}$ with corresponding action set $\mathcal{A}$ and prediction $\hat{p}$, define the Bayes decision function as $\delta_\ell(\hat{p}(x)) = \arg\inf_{a \in \mathcal{A}} \mathbb{E}_{\hat{Y} \sim \hat{p}(x)}[\ell(\hat{Y}, a)]$. For any subset $\mathcal{L} \subset \mathcal{L}_{\text{all}}$ denote the set of all Bayes decision functions as $\mathbb{A}_{\mathcal{L}} := \{\delta_\ell \mid \forall \ell \in \mathcal{L}\}$.*

# 3 Calibration: A Decision-Making Perspective

In our setup, the decision-maker outsources the prediction task to a third-party forecaster, which returns a prediction function $\hat{p}$ (e.g., an ML Prediction API). The decision maker will use $\hat{p}$ and a loss function $\ell$ to make decisions. However, the forecaster does not necessarily know the loss function $\ell$ in advance, and in more challenging cases, needs to serve multiple decision-makers with different loss functions. For example, the prediction function may be trained to optimize a standard objective such as L2 error or log likelihood, then sold to decision-makers as an off-the-shelf solution.

In such a setting, the decision makers may be concerned that the off-the-shelf solution may not perform well according to their loss function. If the forecaster could predict optimally (i.e. $\hat{p}(X) = p^*(X)$ almost surely), then there would be no issue of trust; of course, perfect predictions are usually impossible, so the forecaster needs feasible ways of conveying trust to the decision makers. To mitigate concerns about the performance of the prediction function, the forecaster might aim to offer performance guarantees applicable to decision makers whose loss functions come from class of losses $\mathcal{L} \subset \mathcal{L}_{\text{all}}$.

## 3.1 Decision Calibration

First and foremost, a decision maker wants assurance that the Bayes decision rule $\delta_\ell$ with $\hat{p}$ gives low expected loss. Second, the decision maker wants to know how much loss is going to be incurred (before the actions are deployed and outcomes are revealed); the decision maker does not want to incur any additional loss in surprise that she has not prepared for.

To capture these desiderata we formalize a definition based on the following intuition: suppose a decision maker with some loss function $\ell$ considers a decision rule $\delta \in \mathbb{A}_{\text{all}}$ (that may or may not be the Bayes decision rule), the decision maker should be able to correctly compute the expected loss of using $\delta$ to make decisions, as a function of *the predictions $\hat{p}$*.

**Definition 2** (Decision Calibration). *For a set of loss functions $\mathcal{L} \subset \mathcal{L}_{\text{all}}$ and a set of decision rules $\mathbb{A} \subset \mathbb{A}_{\text{all}}$, we say that a prediction $\hat{p}$ is $(\mathcal{L}; \mathbb{A})$-decision calibrated (with respect to $p^*$) if $\forall \ell \in \mathcal{L}$ and $\delta \in \mathbb{A}$ with the same action space $\mathcal{A}$ [2], the computed loss (based on $\hat{p}$) of $\delta$ is the same as the actual*

---

[2]We require $\ell$ and $\delta$ to have the same action space $\mathcal{A}$ for type check reasons. Eq.(2) only has meaning if the loss $\ell$ and decision rule $\delta$ are associated with the same action space $\mathcal{A}$.

*loss, i.e.*

$$\mathbb{E}_X \mathbb{E}_{\hat{Y} \sim \hat{p}(X)}[\ell(\hat{Y}, \delta(\hat{p}(X)))] = \mathbb{E}_X \mathbb{E}_{Y \sim p^*(X)}[\ell(Y, \delta(\hat{p}(X)))] \quad (1)$$

*In particular, we say $\hat{p}$ is $\mathcal{L}$-decision calibrated if it is $(\mathcal{L}; \triangle_{\mathcal{L}})$-decision calibrated, where $\triangle_{\mathcal{L}}$ is the set of all Bayes decision rules for loss functions in $\mathcal{L}$.*

The left hand side of Eq.(2) is the "simulated" loss where the outcome $\hat{Y}$ is hypothetically drawn from the predicted distribution. The decision maker can compute this just by knowing the input features $X$ and without knowing the outcome $Y$. The right hand side of Eq.(2) is the true loss that the decision maker incurs in reality if she uses the decision rule $\delta$. Intuitively, the definition captures the idea that the losses in $\mathcal{L}$ and decision rules in $\triangle$ do not distinguish between outcomes sampled according to the predicted probability and the true probability; specifically, the definition can be viewed as an instantiation of the framework of *Outcome Indistinguishability* (Dwork et al., 2021).

As a cautionary remark, Eq.(2) should not be mis-interpreted as guarantees about individual decisions; Eq.(2) only looks at the average loss when $X, Y$ is a random draw from the population. Individual guarantees are usually impossible without tools beyond machine learning (Zhao & Ermon, 2021). In addition, Definition 2 does not consider decision rules that can directly depend on $X$, as $\delta \in \triangle_{\text{all}}$ only depends on $X$ via the predicted probability $\hat{p}(X)$. Studying decision rules that can directly depend on $X$ require tools such as multicalibration (Hébert-Johnson et al., 2018) which are beyond the scope of this paper (see related work).

In Definition 2 we also define the special notion of $\mathcal{L}$-decision calibrated because given a set of loss functions $\mathcal{L}$, we are often only interested in the associated Bayes decision rules $\triangle_{\mathcal{L}}$, i.e. the set of decision rules that are optimal under *some* loss function. For the rest of the paper we focus on $\mathcal{L}$-decision calibration for simplicity. $\mathcal{L}$-decision calibration can capture the desiderata we discussed above formalized in the following proposition.

**Proposition 1.** *If a prediction function $\hat{p}$ is $\mathcal{L}$-decision calibrated, then it satisfies $\forall \delta' \in \triangle_{\mathcal{L}}$*

$$\mathbb{E}_X \mathbb{E}_{Y \sim p^*(X)}[\ell(Y, \delta_\ell(\hat{p}(X)))] \leq \mathbb{E}_X \mathbb{E}_{Y \sim p^*(X)}[\ell(Y, \delta'(\hat{p}(X)))] \quad \textit{(Bayes Decision Optimality)}$$

$$\mathbb{E}_X \mathbb{E}_{\hat{Y} \sim \hat{p}(X)}[\ell(\hat{Y}, \delta_\ell(\hat{p}(X)))] = \mathbb{E}_X \mathbb{E}_{Y \sim p^*(X)}[\ell(Y, \delta_\ell(\hat{p}(X)))] \quad \textit{(Accurate loss estimation)}$$

**Bayes Decision Optimality** states that the Bayes decision rule $\delta_\ell$ is not worse than any alternative decision rule $\delta' \in \triangle_{\mathcal{L}}$. In other words, the decision maker is incentivized to take optimal actions according to their true loss function. That is, using the predictions given by $\hat{p}$, the decision maker cannot improve their actions by using a decision rule $\delta'$ that arises from a different loss function $\ell' \in \mathcal{L}$.

**Accurate loss estimation** states that for the Bayes decision rule $\delta_\ell$, the simulated loss on the left hand side (which the decision maker can compute before the outcomes are revealed) equals the true loss on the right hand side. This ensures that the decision maker knows the expected loss that will be incurred over the distribution of individuals and can prepare for it. This is important because in most prediction tasks, the labels $Y$ are revealed with a significant delay or never revealed. For example, the hospital might be unable to follow up on the true outcome of all of its patients.

In practice, the forecaster chooses some set $\mathcal{L}$ to achieve $\mathcal{L}$-decision calibration, and advertise it to decision makers. A decision makers can then check whether their loss function $\ell$ belongs to the advertised set $\mathcal{L}$. If it does, the decision maker should be confident that the Bayes decision rule $\delta_\ell$ has low loss compared to alternatives in $\triangle_{\mathcal{L}}$, and they can know in advance the loss that will be incurred.

### 3.2 Decision Calibration Generalizes Existing Notions of Calibration

We show that by varying the choice of loss class $\mathcal{L}$, decision calibration can actually express prior notions of calibration. For example, consider confidence calibration, where among the samples whose the top probability is $\beta$, the top accuracy is indeed $\beta$. Formally, confidence calibration requires that

$$\Pr[Y = \arg\max \hat{p}(X) \mid \max \hat{p}(X) = \beta] = \beta.$$

We show that a prediction function $\hat{p}$ is confidence calibrated if and only if it is $\mathcal{L}_r$-decision calibration, where $\mathcal{L}_r$ is defined by

$$\mathcal{L}_r := \{\ell(y, a) = \mathbb{I}(y \neq a \wedge a \neq \bot) + \beta \cdot \mathbb{I}(a = \bot) \mid a \in \mathcal{Y} \cup \{\bot\}, \forall \beta \in [0, 1]\}$$

| Existing Calibration Definitions | Associated Loss Functions |
|---|---|
| Confidence Calibration (Guo et al., 2017) $\Pr[Y = \arg\max \hat{p}(X) \mid \max \hat{p}(X) = \beta] = \beta$ $\forall \beta \in [0,1]$ | $\mathcal{L}_r := \{\ell(y,a;\beta) \mid \forall \beta \in [0,1]\}$ where $\ell(y,a;\beta) := \mathbb{I}(y \neq a \wedge a \neq \perp) + \beta \cdot \mathbb{I}(a = \perp)$ $a \in \mathcal{Y} \cup \{\perp\}$ |
| Classwise Calibration (Kull et al., 2019) $\mathbb{E}[Y_c \mid \hat{p}_c(X) = \beta] = \beta, \forall c \in [C], \forall \beta \in [0,1]$ | $\mathcal{L}_{cr} := \{\ell_c(y,a;\beta_1,\beta_2,c) \mid \forall \beta_1,\beta_2 \in \mathbb{R}, c \in [C]\}$ where $\ell(y,a;\beta_1,\beta_2,c) := \mathbb{I}(a = \perp) + \beta_1 \cdot \mathbb{I}(y = c \wedge a = T)$ $+ \beta_2 \cdot \mathbb{I}(y \neq c \wedge a = F), \qquad a \in \{T,F,\perp\}$ |
| Distribution Calibration (Kull & Flach, 2015) $\mathbb{E}[Y \mid \hat{p}(X) = q] = q, \forall q \in \Delta^C$ | $\mathcal{L}_{\text{all}} = \{\ell \mid \forall \text{ set } \mathcal{A} \text{ and function } \ell : \mathcal{Y} \times \mathcal{A} \to \mathbb{R}\}$ |

Table 1: A prediction function $\hat{p}$ satisfies the calibration definitions on the left if and only if it satisfies $\mathcal{L}$-decision calibration for the loss function families on the right (Theorem 1).

Intuitively, loss functions in $\mathcal{L}_r$ corresponds to the refrained prediction task: a decision maker chooses between reporting a class label, or reporting "I don't know," denoted $\perp$. She incurs a loss of 0 for correctly predicting the label $y$, a loss of 1 for reporting an incorrect class label, and a loss of $\beta < 1$ for reporting "I don't know". If a decision maker's loss function belong to this simple class of losses $\mathcal{L}_r$, he or she can use a confidence calibrated prediction function $\hat{p}$, because the two desiderata (Bayes decision optimality and accurate loss estimation) in Proposition 1 are true for the decision maker. However, such "refrained prediction" decision tasks only account for a tiny subset of all possible tasks that are interesting to decision makers. Similarly, classwise calibration can be characterized through decision calibration using a class of loss functions that penalizes class-specific false positives and negatives.

In this way, decision calibration clarifies the implications of existing notions of calibration on decision making: relaxed notions of calibration correspond to decision calibration over restricted classes of losses. In general, decision calibration provides a unified view of most existing notions of calibration as the following theorem shows.

**Theorem 1.** *[Decision Calibration Generalizes Existing Notions] For any true distribution $p^*$, and for the loss function sets $\mathcal{L}_r, \mathcal{L}_{cr}$ defined in Table 1, a prediction function $\hat{p}$ is*

- *confidence calibrated iff it is $\mathcal{L}_r$-decision calibrated.*

- *classwise calibrated iff it is $\mathcal{L}_{cr}$-decision calibrated.*

- *distribution calibrated iff it is $\mathcal{L}_{all}$-decision calibrated.*

For proof of this theorem see Appendix C. In Table 1, confidence and classwise calibration are weak notions of calibration; correspondingly the loss function families $\mathcal{L}_r$ and $\mathcal{L}_{cr}$ are also very restricted.

On the other hand, distribution calibration (i.e. $\mathbb{E}[Y \mid \hat{p}(X) = q] = q, \forall q \in \Delta^C$) is equivalent to $\mathcal{L}_{\text{all}}$-decision calibration. This means that a distribution calibrated predictor guarantees the Bayes decision optimality and accurate loss estimation properties as in Proposition 1 to a decision maker holding any loss functions. Unfortunately, distribution calibration is very challenging to verify or achieve. To understand the challenges, consider certifying whether a given predictor $\hat{p}$ is distribution calibrated. Because we need to reason about the conditional distribution $\mathbb{E}[Y \mid \hat{p}(X) = q]$ for every $q$ that $\hat{p}$ can predict (i.e. the support of $\hat{p}$), the necessary sample complexity grows linearly in the support of $\hat{p}$. Of course, for a trivial predictors that map all inputs $x$ to the same prediction $q_0$ (i.e. $\hat{p}(x) = q_0, \forall x \in \mathcal{X}$) distribution calibration is easy to certify (Widmann et al., 2019), but such predictors have no practical use.

Our characterization of distribution calibration further sheds light on why it is so difficult to achieve. $\mathcal{L}_{\text{all}}$ consists of *all* loss function, including all loss functions $\ell : \mathcal{Y} \times \mathcal{A} \to \mathbb{R}$ whose action space $\mathcal{A}$ contains exponentially many elements (e.g. $|\mathcal{A}| = 2^C$). The corresponding decision rules $\delta \in \Delta_{\mathcal{L}_{\text{all}}} : \Delta^C \to \mathcal{A}$ may also map $\Delta^C$ to exponentially many possible values. Enforcing Definition 2 for such complex loss functions and decision rules is naturally difficult.

### 3.3 Decision Calibration over Bounded Action Space

In many contexts, directly optimizing for distribution calibration may be overkill. In particular, in most realistic settings, decision makers tend to have a bounded number of possible actions, so the relevant losses come from $\mathcal{L}^K$ for reasonable $K \in \mathbb{N}$. Thus, we consider obtaining decision calibration for all loss functions defined over a bounded number of actions $K$. In the remainder of the

paper, we focus on this restriction of decision calibration to the class of losses with bounded action space; we reiterate the definition of decision calibration for the special case of $\mathcal{L}^K$.

**Definition 3** ($\mathcal{L}^K$-Decision Calibration). *Let $\mathcal{L}^K$ be the set of all loss functions with $K$ actions $\mathcal{L}^K = \{\ell \mid \forall \mathcal{A}, |\mathcal{A}| = K, \forall \ell : \mathcal{Y} \times \mathcal{A} \to \mathbb{R}\}$, we say that a prediction $\hat{p}$ is $\mathcal{L}^K$-decision calibrated (with respect to $p^*$) if $\forall \ell \in \mathcal{L}^K$ and $\delta \in \mathbb{A}_{\mathcal{L}^K}$*

$$\mathbb{E}_X \mathbb{E}_{\hat{Y} \sim \hat{p}(X)}[\ell(\hat{Y}, \delta(\hat{p}(X)))] = \mathbb{E}_X \mathbb{E}_{Y \sim p^*(X)}[\ell(Y, \delta(\hat{p}(X)))] \tag{2}$$

The key result (that we show in Section 4) is that $\mathcal{L}^K$-decision calibration can be efficiently verified and achieved for *all* predictors. Specifically, we show that the sample complexity necessary to learn $\mathcal{L}^K$-decision calibrated predictors depends on the number of actions $K$, not on the support of $\hat{p}$. This is in contrast to the standard approach for establishing distribution calibration, where the sample complexity scales with the support of $\hat{p}$ (which is typically exponential in the number of classes).

# 4 Achieving Decision Calibration with PAC Guarantees

## 4.1 Approximate $\mathcal{L}^K$ Decision Calibration is Verifiable and Achievable

Decision calibration in Definition 2 usually cannot be achieved perfectly. The definition has to be relaxed to allow for statistical and numerical errors. To meaningfully define approximate calibration we assume that the loss functions are bounded, i.e. no outcome $y \in \mathcal{Y}$ and action $a \in \mathcal{A}$ can incur an infinite loss. In particular, we bound $\ell$ by its 2-norm $\max_a \|\ell(\cdot, a)\|_2 := \max_a \sqrt{\sum_{y \in \mathcal{Y}} \ell(y, a)^2}$. [3]

Now we can proceed to define approximate decision calibration. In particular, we compare the difference between the two sides in Eq.(2) of Definition 2 with the maximum magnitude of the loss function.

**Definition 4** (Approximate decision calibration). *A prediction function $\hat{p}$ is $(\mathcal{L}, \epsilon)$-decision calibrated (with respect to $p^*$) if $\forall \ell \in \mathcal{L}$ and $\delta \in \mathbb{A}_{\mathcal{L}}$*

$$\left| \mathbb{E}[\ell(\hat{Y}, \delta(\hat{p}(X)))] - \mathbb{E}[\ell(Y, \delta(\hat{p}(X)))] \right| \leq \epsilon \sup_{a \in \mathcal{A}} \|\ell(\cdot, a)\|_2 \tag{3}$$

Definition 4 is a relaxation of Definition 2: if a prediction function is $(\mathcal{L}, 0)$-decision calibrated, then it is $\mathcal{L}$-decision calibrated (Definition 2).

The main observation in our paper is that for the set of loss functions with $K$ actions, $(\mathcal{L}^K, \epsilon)$-decision calibration can be verified and achieved with polynomially many samples. In addition, we achieve decision calibration without deteriorating the $L_2$ error $\mathbb{E}[\|\hat{p}(X) - Y\|_2^2]$.

**Theorem 2** (Main Theorem, informal). *There is an algorithm, such that for any predictor $\hat{p}$ and given polynomially (in $K, C, 1/\epsilon$) many samples, can with high probability*

1. *correctly decide if $\hat{p}$ satisfies $(\mathcal{L}^K, \epsilon)$-decision calibration*

2. *output a new predictor $\hat{p}'$ that satisfies $(\mathcal{L}^K, \epsilon)$-decision calibration without degrading the original predictions (in terms of $L_2$ error).*

Note that trivial predictors $\hat{p}(x) \equiv \mathbb{E}[Y], \forall x$ satisfy $(\mathcal{L}^K, 0)$-decision calibration. To maintain a meaningful prediction function we also require "sharpness" which we measure by $L_2$ error. We guarantee that the $L_2$ error of $\hat{p}'$ can only *improve* under post-processing; that is, $\mathbb{E}[\|\hat{p}'(X) - Y\|_2^2] \leq \mathbb{E}[\|\hat{p}(X) - Y\|_2^2]$. In fact, the algorithm works by iteratively updating the predictions to make progress in $L_2$. In addition to $L_2$ error, empirically our recalibration algorithm also slightly improves the likelihood and accuracy. For rest of this section, we propose concrete algorithms that satisfy Theorem 2.

## 4.2 Verification of Decision Calibration

This section focuses on the first part of Theorem 2 where we certify $(\mathcal{L}^K, \epsilon)$-decision calibration. A naive approach use samples to directly estimate (by replacing all expectations with empirical sample

---

[3]The choice of 2-norm is for convenience. All $p$-norms are equivalent up to a multiplicative factor polynomial in $C$, so our main theorem (Theorem 2) still hold for any $p$-norms up to the polynomial factor.

averages)

$$\sup_{\delta \in \mathbb{A}_{\mathcal{L}^K}} \sup_{\ell \in \mathcal{L}^K} \left| \mathbb{E}[\ell(\hat{Y}, \delta(\hat{p}(X)))] - \mathbb{E}[\ell(Y, \delta(\hat{p}(X)))] \right| / \left( \sup_{a \in \mathcal{A}} \|\ell(\cdot, a)\|_2 \right) \tag{4}$$

and compare it with $\epsilon$. Even though it might be possible to upper bound the estimation error for Eq.(4) directly, the analysis would be complex because of the multiple sup in the equation. To gain deeper insight into the problem and simplify its analysis, we will make several observations to transform this complex optimization problem to a simpler problem that resembles linear classification.

**Observation 1.** The first observation is that we do not have to take the supremum over $\mathcal{L}^K$ because for any choice of $\delta \in \mathbb{A}$ by simple calculations we have

$$\sup_{\ell \in \mathcal{L}^K} \frac{\left| \mathbb{E}[\ell(\hat{Y}, \delta(\hat{p}(X)))] - \mathbb{E}[\ell(Y, \delta(\hat{p}(X)))] \right|}{\sup_{a \in \mathcal{A}} \|\ell(\cdot, a)\|_2} = \sum_{a=1}^{K} \left\| \mathbb{E}[(\hat{Y} - Y) \mathbb{I}(\delta(\hat{p}(X)) = a)] \right\|_2 \tag{5}$$

This statement is formally proved as part of the proof for Proposition 3 in Appendix C.2. Intuitively on the right hand side, $\delta$ partitions the probabilities $\Delta^C$ based on the optimal decision $\Delta_1 := \{q \in \Delta^C, \mathbb{I}(\delta(q) = 1)\}, \cdots, \Delta_K := \{q \in \Delta^C, \mathbb{I}(\delta(q) = K)\}$. For each partition $\Delta_k$ we measure the difference between predicted label and true label *on average*, i.e. $\mathbb{E}[(\hat{Y} - Y)\mathbb{I}(\hat{p}(X) \in \Delta_k)]$.

**Observation 2.** We observe that the partitions of $\Delta^C$ are defined by linear classification boundaries. Formally, we introduce a new notation for the linear multi-class classification functions

$$B^K = \left\{ b_w \mid \forall w \in \mathbb{R}^{K \times C} \right\} \qquad \text{where } b_w(q, a) = \mathbb{I}(a = \arg \sup_{a' \in [K]} \langle q, w_{a'} \rangle) \tag{6}$$

Note that this new classification task is a tool to aid in understanding decision calibration, and bears no relationship with the original prediction task (predicting $Y$ from $X$). Intuitively $w$ defines the weights of a linear classifier; given input features $q \in \Delta^C$ and a candidate class $a$, $b_w$ outputs an indicator: $b_w(q, a) = 1$ if the optimal decision of $q$ is equal to $a$ and 0 otherwise.

The following equality draws the connection between Eq.(5) and linear classification. The proof is simply a translation from the original notations to the new notations.

$$\sup_{\delta \in \mathbb{A}_{\mathcal{L}^K}} \sum_{a=1}^{K} \left\| \mathbb{E}[(\hat{Y} - Y) \mathbb{I}(\delta(\hat{p}(X)) = a)] \right\|_2 = \sup_{b \in B^K} \sum_{a=1}^{K} \left\| \mathbb{E}[(\hat{Y} - Y) b(\hat{p}(X), a)] \right\|_2 \tag{7}$$

The final outcome of our derivations is the following proposition (Proof in Appendix C.2)

**Proposition 2.** *A predictor $\hat{p}$ satisfies $(\mathcal{L}^K, \epsilon)$-decision calibration if and only if*

$$\sup_{b \in B^K} \sum_{a=1}^{K} \left\| \mathbb{E}[(\hat{Y} - Y) b(\hat{p}(X), a)] \right\|_2 \leq \epsilon \tag{8}$$

In words, $\hat{p}$ satisfies decision calibration if and only if there is no linear classification function that can partition $\Delta^C$ into $K$ parts, such that the average difference $\hat{Y} - Y$ (or equivalently $\hat{p}(X) - p^*(X)$) in each partition is large. Theorem 2.1 follows naturally because $B^K$ has low Radamacher complexity, so the left hand side of Eq.(8) can be accurately estimated with polynomially many samples. For a formal statement and proof see Appendix C.2.

The remaining problem is computation. With unlimited compute, we can upper bound Eq.(8) by brute force search over $B^K$; in practice, we use a surrogate objective optimizable with gradient descent. This is the topic of Section 4.4.

### 4.3 Recalibration Algorithm

This section discusses the second part of Theorem 2 where we design a post-processing recalibration algorithm. The algorithm is based on the following intuition, inspired by (Hébert-Johnson et al., 2018): given a predictor $\hat{p}$ we find the worst $b \in B^K$ that violates Eq.(8) (line 3 of Algorithm 1); then,

we make an update to $\hat{p}$ to minimize the violation of Eq.(8) for the worst $b$ (line 4,5 of Algorithm 1). This process is be repeated until we get a $(B^K, \epsilon)$-decision calibrated prediction (line 2). The sketch of the algorithm is shown in Algorithm 1 and the detailed algorithm is in the Appendix.

---

**Algorithm 1:** Recalibration algorithm to achieve $\mathcal{L}^K$ decision calibration.

1   Input current prediction function $\hat{p}$, tolerance $\epsilon$. Initialize $\hat{p}^{(0)} = \hat{p}$;
2   **for** $t = 1, 2, \cdots, T$ *until output* $\hat{p}^{(T)}$ *when it satisfies Eq.(8)* **do**
3      Find $b \in B^K$ that maximizes $\sum_{a=1}^{K} \left\| \mathbb{E}[(Y - \hat{p}^{(t-1)}(X))b(\hat{p}^{(t-1)}(X), a)] \right\|$ ;
4      Compute the adjustments $d_a = \mathbb{E}[(Y - \hat{p}^{(t-1)}(X))b(\hat{p}^{(t-1)}(X), a)]/\mathbb{E}[b(\hat{p}^{(t-1)}(X), a)]$ ;
5      Set $\hat{p}^{(t)} : x \mapsto \hat{p}^{(t-1)}(x) + \sum_{a=1}^{K} b(\hat{p}^{(t-1)}(x), a) \cdot d_a$ (projecting onto $[0, 1]$ if necessary) ;
6   **end**

---

Given a dataset with $N$ samples, the expectations in Algorithm 1 are replaced with empirical averages. The following theorem demonstrates that Algorithm 1 satisfies the conditions stated in Theorem 2.

**Theorem 2.2.** *Given any input $\hat{p}$ and tolerance $\epsilon$, Algorithm 1 terminates in $O(K/\epsilon^2)$ iterations. For any $\lambda > 0$, given $O(poly(K, C, \log(1/\delta), \lambda))$ samples, with $1 - \delta$ probability Algorithm 1 outputs a $(\mathcal{L}^K, \epsilon + \lambda)$-decision calibrated prediction function $\hat{p}'$ that satisfies $\mathbb{E}[\|\hat{p}'(X) - Y\|_2^2] \leq \mathbb{E}[\|\hat{p}(X) - Y\|_2^2] + \lambda$.*

The theorem demonstrates that if we can solve the inner search problem over $B^K$, then we can obtain decision calibrated predictions efficiently in samples and computation.

### 4.4 Relaxation of Decision Calibration for Computational Efficiency

We complete the discussion by addressing the open computational question. Directly optimizing over $B^K$ is difficult, so we instead define the softmax relaxation.

$$\bar{B}^K = \left\{ \bar{b}_w \mid \forall w \in \mathbb{R}^{K \times C} \right\} \qquad \text{where } \bar{b}_w(q, a) = \frac{e^{\langle q, w_a \rangle}}{\sum_{a'} e^{\langle q, w_{a'} \rangle}}$$

The key motivation behind this relaxation is that $\bar{b}_w \in \bar{B}^K$ is now differentiable in $w$, so we can optimize over $\bar{B}^K$ using gradient descent. Correspondingly some technical details in Algorithm 1 change to accommodate soft partitions; we address these modifications in Appendix A and show that after these modifications Theorem 2.2 still holds. Intuitively, the main reason that we can still achieve decision calibration with softmax relaxation is because $B^K$ is a subset of the closure of $\bar{B}^K$. Therefore, compared to Eq.(8), we enforce a slightly stronger condition with the softmax relaxation. This can be formalized in the following proposition.

**Proposition 3.** *A prediction function $\hat{p}$ is $(\mathcal{L}^K, \epsilon)$-decision calibrated if it satisfies*

$$\sup_{\bar{b} \in \bar{B}^K} \sum_{a=1}^{K} \left\| \mathbb{E}[(\hat{Y} - Y)\bar{b}(\hat{p}(X), a)] \right\|_2 \leq \epsilon : \tag{9}$$

We remark that unlike Proposition 2 (which is an "if and only if" statement), Proposition 3 is a "if" statement. This is because Eq.(9) implies Eq.(8) but not vice versa.

## 5 Empirical Evaluation

### 5.1 Skin Legion Classification

This experiment materializes our motivating example in the introduction. We aim to show on a real medical prediction dataset, our recalibration algorithm improves both the decision loss and reduces the decision loss estimation error. For the estimation error, as in Definition 4 for any loss function $\ell$ and corresponding Bayes decision rule $\delta_\ell$ we measure

$$\text{loss gap} := \left| \mathbb{E}[\ell(\hat{Y}, \delta_\ell(\hat{p}(X)))] - \mathbb{E}[\ell(Y, \delta_\ell(\hat{p}(X)))] \right| / \sup_{a \in \mathcal{A}} \|\ell(\cdot, a)\|_2 \tag{10}$$

In addition to the loss function in Figure 1 (which is motivated by medical domain knowledge), we also consider a set of 500 random loss functions where for each $y \in \mathcal{Y}, a \in \mathcal{A}, \ell(y, a) \sim \text{Normal}(0, 1)$, and report both the average loss gap and the maximum loss gap across the loss functions.

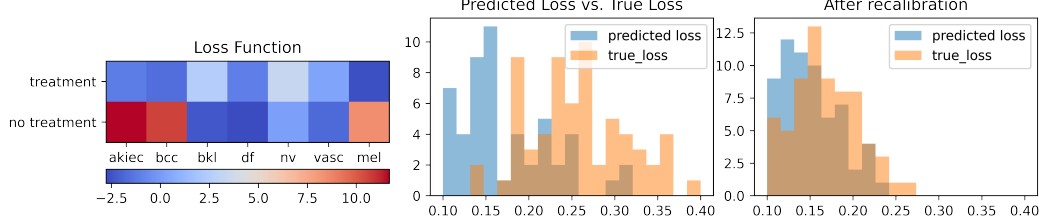

Figure 1: An illustrative example of accurate loss estimation on the HAM10000 dataset. The predictor predicts the probability of 7 skin illness categories (akiec, bcc, ...., mel); the hospital decides between two actions (treatment vs. no treatment). **Left** An example loss function (details not important). Blue indicates low loss and red indicates high loss. For the malignant conditions such as 'akiec' and 'mel', no treatment has high loss (red); for the benign conditions such as 'nv', (unnecessary) treatment has moderate loss. **Middle** Histogram of predicted loss vs. the true loss. The predicted loss is the loss the hospital expected to incur assuming the predicted probabilities are correct under the Bayes decision rule.The true loss is the loss the hospital actually incurs (after ground-truth outcomes are revealed). We plot the histogram from random train/test splits of the dataset, and observe that the true loss is generally much greater than the predicted loss. Because the hospital might incur loss it didn't expect or prepare for, the hospital cannot trust the prediction function. **Right** After applying our recalibration algorithm, the true loss almost matches the predicted loss.

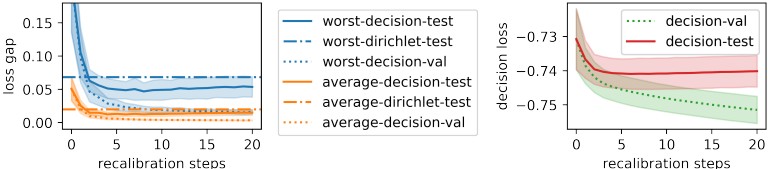

Figure 2: Calibration improves decision loss and its estimation on the HAM10000 skin legion classification dataset. **Left** The gap between the predicted decision loss and the true decision loss in Eq.(10) on a set of randomly sampled loss functions. We plot both the average gap (orange) and the worst gap (blue) out of all the loss functions. The dotted lines are the validation set performance, and solid lines are the test set performance. With more recalibration steps in Algorithm 1, both the average gap and the worst gap improves. The improvements are greater than Dirichlet recalibration (dashed line) **Right** The decision loss (averaged across all random loss functions). With more recalibration steps the decision loss also improves slightly. Dirichlet calibration (dash-dot line) also improves the loss gap for this dataset, but not as much as decision calibration. Note that Dirichlet calibration only has to be applied once, so its performance is a constant instead of a function of the recalibration steps.

**Setup** We use the HAM10000 dataset (Tschandl et al., 2018). We partition the dataset into train/validation/test sets, where approximately 15% of the data are used for validation, while 10% are used for the test set. We use the train set to learn the baseline classifier $\hat{p}$, validation set to recalibrate, and the test set to measure final performance. For modeling we use the densenet-121 architecture (Huang et al., 2017), which achieves around 90% accuracy.

For our method we use Algorithm 2 in Appendix A (which is a small extension of Algorithm 1 explained in Section 4.4). We compare with temperature scaling (Guo et al., 2017) and Dirichlet calibration (Kull et al., 2019). In this experiment, we first apply temperature scaling then apply other methods (decision calibration or Dirichlet calibration). The temperature scaling baseline corresponds to 0 decision recalibration steps in Figure 2.

**Results** The results are shown in Figure 2. For these experiments we set the number of actions $K = 3$. For other choices we obtain qualitatively similar results in Appendix B. The main observation is that decision recalibration improves the loss gap in Eq.(10) and slightly improves the decision loss. We also observe that our recalibration algorithm slightly improves top-1 accuracy (the average improvement is $0.40 \pm 0.08\%$) and L2 loss (the average decrease is $0.010 \pm 0.001$) on the test set.

### 5.2 Imagenet Classification

We stress test our algorithm on Imagenet. The aim is to show that even with deeply optimized classifiers (such as inception-v3 and resnet) that are tailor made for the Imagenet benchmark and a large number of classes, our recalibration algorithm can improve the loss gap in Eq.(10).

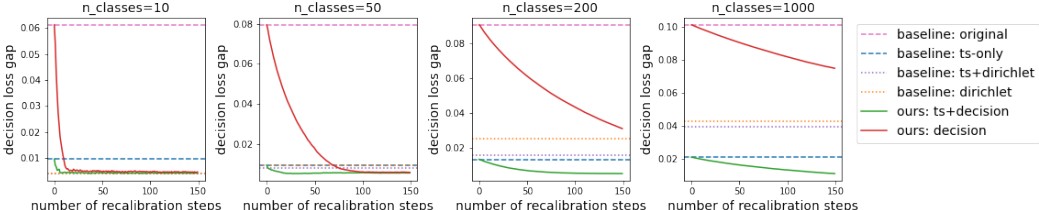

Figure 3: Calibration improves decision loss estimation gap on Imagenet-1000 with inception-v3 on the test set. The meaning of the plot is identical to Figure 2. From left to right we vary the number of classification categories $C$ from 10 to 1000. Our algorithm reduces the gap between predicted loss and true loss in Eq.(10) even up to 1000 classes, although more classes require more iterations and are more prone to over-fitting. Dirichlet calibration is less scalable than our method, and has worse loss estimation with more classes. Using temperature scaling (ts) in additional to decision calibration drastically reduces the number of iterations required for decision calibration to converge.

**Setup** The setup and baselines are identical to the HAM10000 experiment with two differences: we use pretrained models provided by pytorch, and among the 50000 standard validation samples, we use 40000 samples for recalibration and 10000 samples for testing. Similar to the previous experiment, we randomly generate a set of 500 loss functions from normal distributions.

We additional test the performance of decision calibration under distribution drift on the Imagenet-C benchmark (Hendrycks & Dietterich, 2019). We partition the Imagenet validation data into a recalibration set and a test set. We train decision calibration on the recalibration set (with no corruption), and measure performance on the test set (with corruption).

**Results** The results are shown in Figure 3 with additional plots in Appendix B. Decision calibration can generalize to a larger number of classes, and still provides some (albeit smaller) benefits with 1000 classes. Recalibration does not hurt accuracy and L2 error, as we observe that both improve by a modest amount (on average by 0.30% and 0.00173 respectively). We contrast decision calibration with Dirichlet calibration (Kull et al., 2019). Dirichlet calibration also reduces the loss gap when the number of classes is small (e.g. 10 classes), but is less scaleble than decision recalibration. With more classes its performance degrades much more than decision calibration. In the appendix Figure 6 and Figure 7, we further show that even under distribution drift (on the Imagenet-C) benchmark, decision calibration can improve both decision loss and decision loss estimation.

# 6 Related Work

**Calibration** Early calibration research focus on binary classification (Brier, 1950; Dawid, 1984). For multiclass classification, the strongest definition is distribution (strong) calibration (Kull & Flach, 2015; Song et al., 2019) but is hindered by sample complexity. Weaker notions such as confidence (weak) calibration (Platt et al., 1999; Guo et al., 2017), class-wise calibration (Kull et al., 2019) or average calibration average calibration (Kuleshov et al., 2018) are more commonly used in practice. To unify these notions, (Widmann et al., 2019) proposes $\mathcal{F}$-calibration but lacks detailed guidance on which notions to use.

**Individual calibration** Our paper discusses the *average* decision loss over the population $X$. A stronger requirement is to guarantee the loss for each individual decision. Usually individual guarantees are near-impossible (Barber et al., 2019) and are only achievable with hedging (Zhao & Ermon, 2021) or randomization (Zhao et al., 2020).

**Multi-calibration and Outcome Indistinguishability.** Calibration have been the focus of many works on fairness, starting with (Kleinberg et al., 2016; Pleiss et al., 2017). Multi-calibration has emerged as a noteworthy notion of fairness (Hébert-Johnson et al., 2018; Kim et al., 2019; Dwork et al., 2019; Shabat et al., 2020; Jung et al., 2020; Dwork et al., 2021) because it goes beyond "protected" groups, and guarantees calibration for any group that is identifiable within some computational bound. Recently, (Dwork et al., 2021) generalizes multicalibration to outcome indistinguishability (OI). Decision calibration is a special form of OI.

# 7 Acknowledgements

SZ is supported in part by a JP Morgan fellowship and a Qualcomm innovation fellowship. MPK is supported by the Miller Institute for Basic Research in Science and the Simons Collaboration on the Theory of Algorithmic Fairness. TM acknowledges support of Google Faculty Award and NSF IIS 2045685. SE acknowledges support by NSF(#1651565, #1522054, #1733686), ONR (N000141912145), AFOSR (FA95501910024), ARO (W911NF-21-1-0125) and Sloan Fellowship.

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
