# A  Relaxations to Decision Calibration

We define the algorithm that corresponds to Algorithm 1 but for softmax relaxed functions. Before defining our algorithm at each iteration $t$ we first lighten our notation with a shorthand $b_a(X) = b(\hat{p}^{(t-1)}(X), a)$ (at different iteration $t$, $b_a$ denotes different functions), and $b(X)$ is the vector of $(b_1(X), \cdots, b_K(X))$.

---

**Algorithm 2:** Recalibration Algorithm to achieve decision calibration.

1  Input current prediction function $\hat{p}$, a dataset $\mathcal{D} = \{(x_1, y_1), \cdots, (x_M, y_M)\}$ tolerance $\epsilon$ ;
2  Initialize $\hat{p}^{(0)} = \hat{p}$, $v^{(0)} = +\infty$;
3  **for** $t = 1, 2, \cdots$ *until* $v^{(t-1)} < \epsilon^2/K$ **do**
4  $\quad$ $v^{(t)}, b^{(t)} = \sup, \arg\sup_{b \in \bar{B}^K} \sum_{a=1}^{K} \left\| \mathbb{E}[(Y - \hat{p}^{(t-1)}(X)) b_a(X)] \right\|^2$ ;
5  $\quad$ Compute $D \in \mathbb{R}^{K \times K}$ where $D_{aa'} = \mathbb{E}[b_a^{(t)}(X) b_{a'}^{(t)}(X)]$ ;
6  $\quad$ Compute $R \in \mathbb{R}^{K \times C}$ where $R_a = \mathbb{E}[(Y - \hat{p}^{(t-1)}(X)) b_a^{(t)}(X)]$ ;
7  $\quad$ Set $\hat{p}^{(t)} : x \mapsto \pi(\hat{p}^{(t-1)}(x) + R^T D^{-1} b^{(t)}(x))$ where $\pi$ is the normalization projection;
8  **end**
9  Output $\hat{p}^{(T)}$ where $T$ is the number of iterations

---

For the intuition of the algorithm, consider the $t$-th iteration where the current prediction function is $\hat{p}^{(t-1)}$. On line 4 we find the worst case $b^{(t)}$ that maximizes the "error" $\sum_{a=1}^{K} \left\| \mathbb{E}[(Y - \hat{p}^{t-1}(X)) b_a^{(t)}(X)] \right\|^2$, and on line 5-7 we make the adjustment $\hat{p}^{(t-1)} \to \hat{p}^{(t)}$ to minimize this error $\sum_{a=1}^{K} \left\| \mathbb{E}[(Y - \hat{p}^t(X)) b_a^{(t)}(X)] \right\|^2$. In particular, the adjustment we aim to find on line 5-7 (which we denote by $U \in \mathbb{R}^{C \times K}$) should satisfy the following: if we let

$$\hat{p}^{(t)}(X) = \hat{p}^{(t-1)}(X) + U b^{(t)}(X)$$

we can minimize

$$L(U) := \sum_{a=1}^{K} \left\| \mathbb{E}[(Y - \hat{p}^{(t)}(X)) b_a^{(t)}(X)] \right\|^2$$

We make some simple algebra manipulations on $L$ to get

$$L(U) = \sum_{a=1}^{K} \left\| \mathbb{E}[(Y - \hat{p}^{(t-1)}(X)) b_a^{(t)}(X) - U b^{(t)}(X) b_a^{(t)}(X)] \right\|^2$$

$$= \sum_{a=1}^{K} \left\| R_a - (D U^T)_a \right\|^2 = \left\| R - D U^T \right\|^2$$

Suppose $D$ is invertible, then the optimum of the objective is

$$U^* := \arg\inf L(U) = R^T D^{-1}, \quad L(U^*) = 0$$

When $D$ is not invertible we can use the pseudo-inverse, though we observe in the experiments that $D$ is always invertible.

For the relaxed algorithm we also have a theorem that is equivalent to Theorem 2.2. The statement of the theorem is identical; the proof is also essentially the same except for the use of some new technical tools.

**Theorem 2.2'.** *Algorithm 2 terminates in $O(K/\epsilon^2)$ iterations. For any $\lambda > 0$, given $O(\text{poly}(K, C, \log(1/\delta), \lambda))$ samples, with $1 - \delta$ probability Algorithm 1 outputs a $(\mathcal{L}^K, \epsilon + \lambda)$-decision calibrated prediction function $\hat{p}'$ that satisfies $\mathbb{E}[\|\hat{p}'(X) - Y\|_2^2] \leq \mathbb{E}[\|\hat{p}(X) - Y\|_2^2] + \lambda$.*

# B  Additional Experiment Details and Results

Additional experiments are shown in Figure 4 and Figure 5. The observations are similar to those in the main paper.

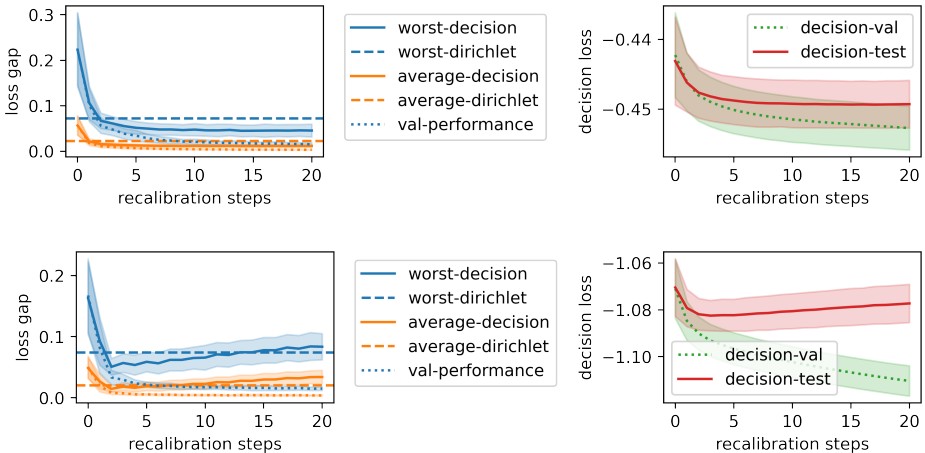

Figure 4: Additional Results on the HAM10000 for 2 and 5 actions. The observations are similar to Figure 2 even though overfitting happens sooner with 5 actions

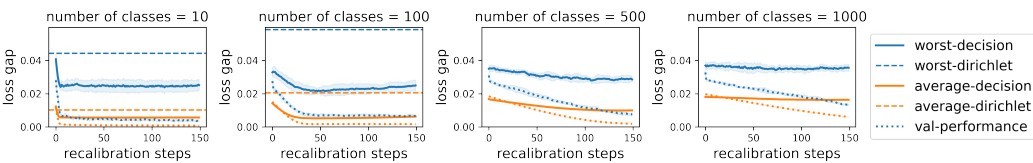

Figure 5: Additional results on the resnet18. The observations are similar to Figure 3: decision recalibration improves the loss gap.

## B.1 Robustness to Distribution Shift

To further test whether decision calibration improves loss under distribution drift, we use the Imagenet-C benchmark (Hendrycks & Dietterich, 2019).

# C Proofs

## C.1 Equivalence between Decision Calibration and Existing Notions of Calibration

**Theorem 1.** *[Decision Calibration Generalizes Existing Notions] For any true distribution $p^*$, and for the loss function sets $\mathcal{L}_r, \mathcal{L}_{cr}$ defined in Table 1, a prediction function $\hat{p}$ is*

- *confidence calibrated iff it is $\mathcal{L}_r$-decision calibrated.*

- *classwise calibrated iff it is $\mathcal{L}_{cr}$-decision calibrated.*

- *distribution calibrated iff it is $\mathcal{L}_{all}$-decision calibrated.*

*Proof of Theorem 1, part 1.* Before the proof we first need a technical Lemma

**Lemma 1.** *For any pair of random variables $U, V$, $\mathbb{E}[U \mid V] = 0$ almost surely if and only if $\forall c \in \mathbb{R}, \mathbb{E}[U\mathbb{1}(V > c)] = 0$.*

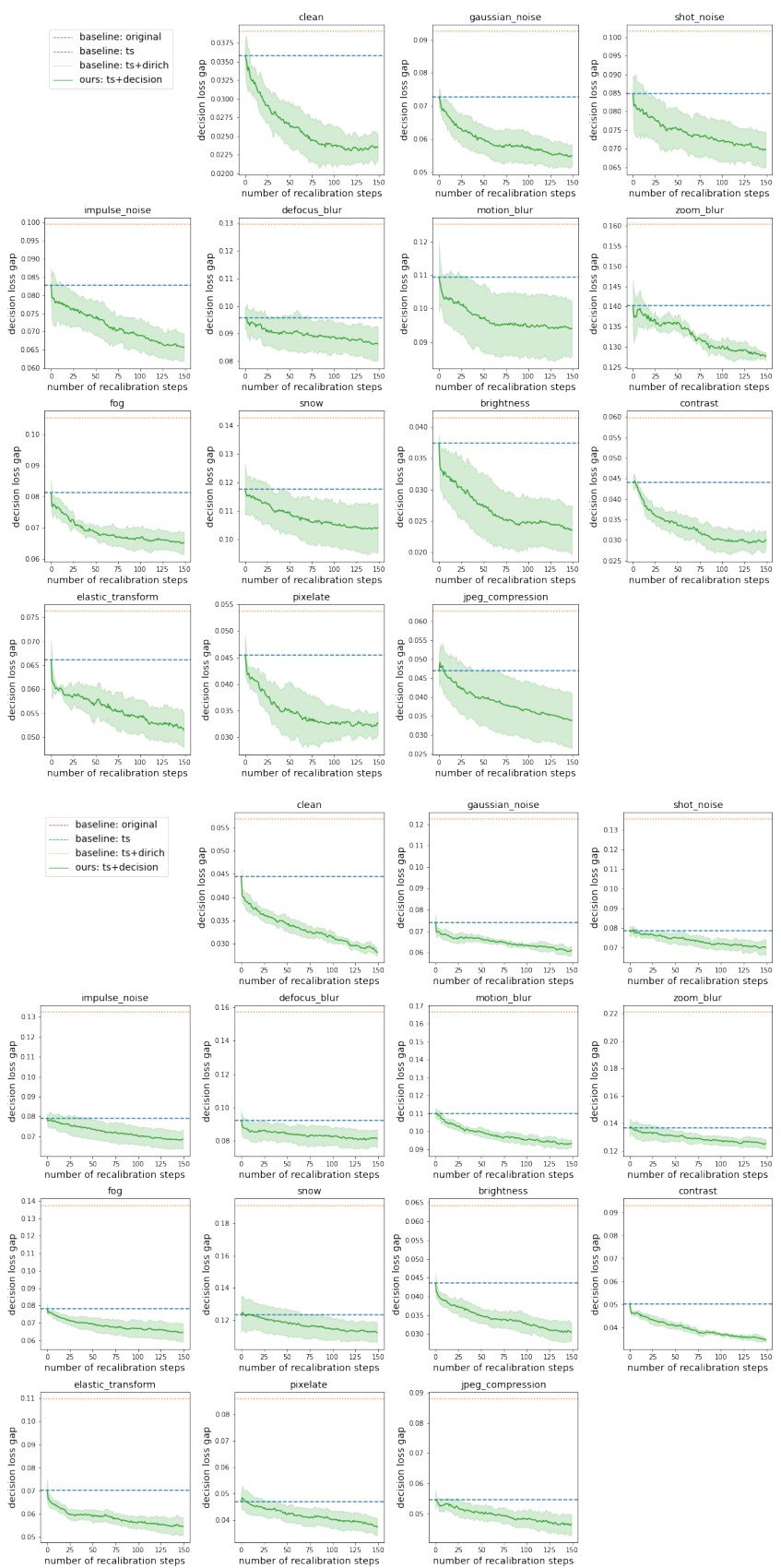

Figure 6: Comparing the maximum difference between predicted loss and actual loss under distribution drifts. The top panel plots the results on 200 classes, and the bottom panel plots the results on 1000 classes.

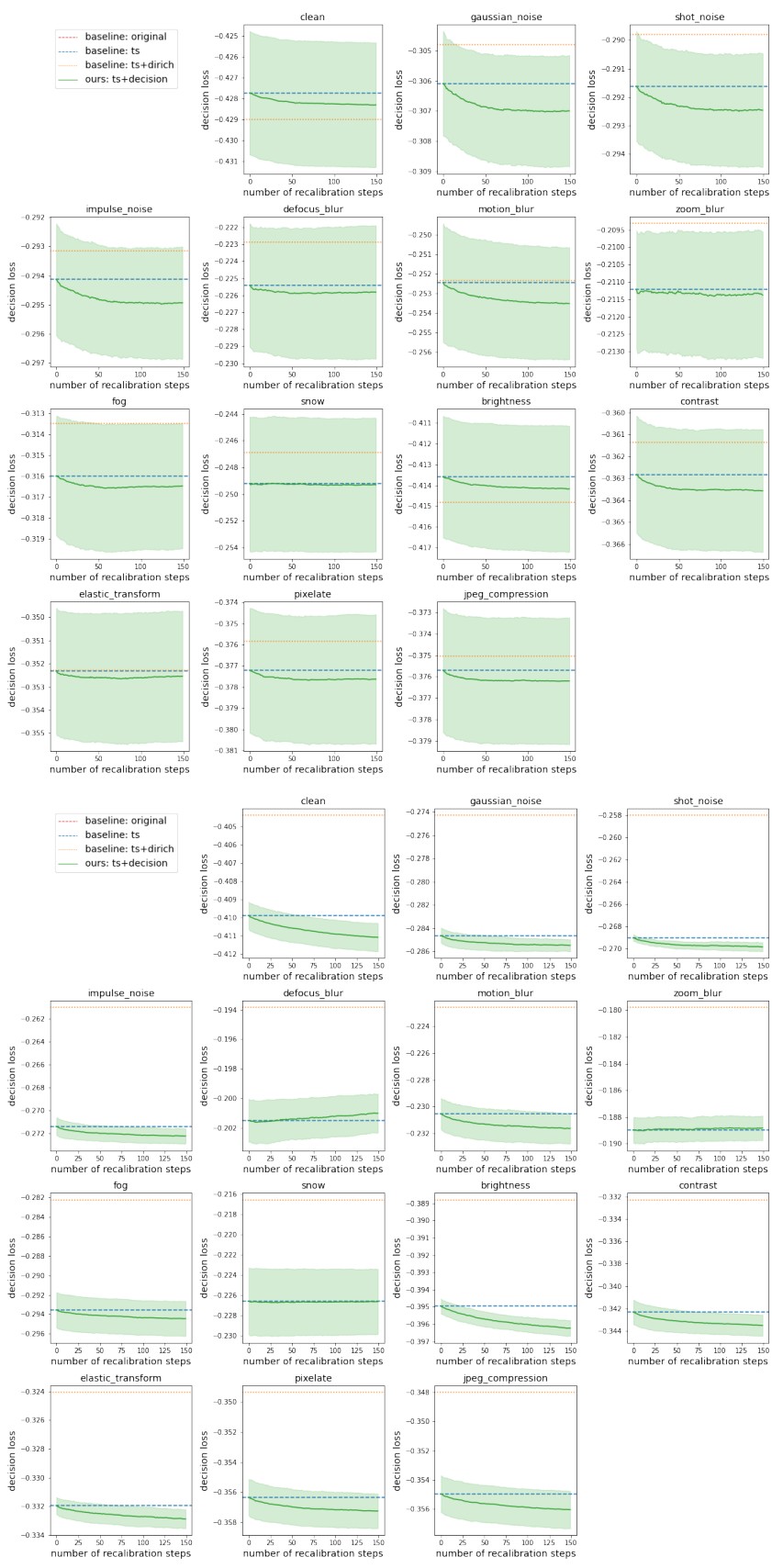

Figure 7: Comparing the decision loss under distribution drifts. The top panel plots the results on 200 classes, and the bottom panel plots the results on 1000 classes.

**Part 1** When the loss function is $\ell : y, a \mapsto \mathbb{I}(y \neq a \cap a \neq \bot) + \beta\mathbb{I}(a = \bot)$, the Bayes decision is given by

$$\delta_\ell(x) = \begin{cases} \arg\max \hat{p}(x) & \max \hat{p}(x) > 1 - \beta \\ \bot & \text{otherwise} \end{cases}$$

Denote $U = \max \hat{p}(X)$ and $V = \arg\max \hat{p}(X)$. For any pair of loss functions $\ell$ and $\ell'$ parameterized by $\beta$ and $\beta'$ we have

$\mathbb{E}[\ell'(Y, \delta_\ell(X))] - \mathbb{E}[\ell'(\hat{Y}, \delta_\ell(X))]$
$= \mathbb{E}[(\ell'(Y, \bot) - \ell'(\hat{Y}, \bot))\mathbb{I}(\delta_\ell(X) = \bot)] + \mathbb{E}[(\ell'(Y, \delta_\ell(X)) - \ell'(\hat{Y}, \delta_\ell(X)))\mathbb{I}(\delta_\ell(X) \neq \bot)]$    Tower
$= 0 + \mathbb{E}[(p_V^*(X) - \hat{p}_V(X))\mathbb{I}(\max \hat{p}(x) > 1 - \beta)]$    Def of $\ell$
$= \mathbb{E}[(p_V^*(X) - U)\mathbb{I}(U > 1 - \beta)]$

Suppose $\hat{p}$ is confidence calibrated, then almost surely

$$U = \Pr[Y = \arg\max \hat{p}(X) \mid U] = \mathbb{E}[p_V^*(X) \mid U]$$

which implies almost surely $\mathbb{E}[p_V^*(X) - U \mid U] = 0$. By Lemma 1 we can conclude that

$$0 = \mathbb{E}[(p_V^*(X) - U)\mathbb{I}(U > 1 - \beta)] = \mathbb{E}[\ell'(Y, \delta_\ell(X))] - \mathbb{E}[\ell'(\hat{Y}, \delta_\ell(X))]$$

so $\hat{p}$ is $L_r$-weakly calibrated.

Conversely suppose $\hat{p}$ is $L_r$ weakly calibrated, then $\forall \beta \in [0, 1]$, $\mathbb{E}[(p_V^*(X) - U)\mathbb{I}(U > 1 - \beta)] = 0$. By Lemma 1 we can conclude that almost surely

$$0 = \mathbb{E}[p_V^*(X) - U \mid U] = \mathbb{E}[p_V^*(X) \mid U] - U$$

so $\hat{p}$ is confidence calibrated.

**Part 2** For any loss function $\ell : y, a \mapsto \mathbb{I}(a = \bot) + \beta_1\mathbb{I}(y \neq c \wedge a = T) + \beta_2\mathbb{I}(y = c \wedge a = F)$ where $\beta_1, \beta_2 > 1$, observe that the Bayes decision for loss function $\ell$ is

$$\delta_\ell(x) = \begin{cases} T & \hat{p}_c(x) > \max(1 - 1/\beta_1, \beta_1/(\beta_1 + \beta_2)) \\ F & \hat{p}_c(x) < \min(1/\beta_2, \beta_1/(\beta_1 + \beta_2)) \\ \bot & \text{otherwise} \end{cases}$$

Choose any pair of numbers $\alpha \geq \gamma$, we can choose $\beta_1, \beta_2$ such that $\alpha := \max(1 - 1/\beta_1, \beta_1/(\beta_1 + \beta_2)), \gamma := \min(1/\beta_2, \beta_1/(\beta_1 + \beta_2))$. For any pair of loss functions $\ell$ and $\ell'$ parameterized by $\beta_1, \beta_2, \beta_1', \beta_2'$ (with associated threshold $\alpha \geq \gamma, \alpha' \geq \gamma'$) we have

$\mathbb{E}[\ell'(Y, \delta_\ell(X)) - \mathbb{E}[\ell'(\hat{Y}, \delta_\ell(X))]$
$= \mathbb{E}[(\ell'(Y, \delta_\ell(X)) - \ell'(\hat{Y}, \delta_\ell(X)))\mathbb{I}(\delta_\ell(X) = T)] + \mathbb{E}[(\ell'(Y, \delta_\ell(X)) - \ell'(\hat{Y}, \delta_\ell(X)))\mathbb{I}(\delta_\ell(X) = F)]$
$= \mathbb{E}[(\ell'(Y, T) - \ell'(\hat{Y}, T))\mathbb{I}(\delta_\ell(X) = T)] + \mathbb{E}[(\ell'(Y, F) - \ell'(\hat{Y}, F))\mathbb{I}(\delta_\ell(X) = F)]$
$= \beta_1'\mathbb{E}[(\hat{p}_c(X) - p_c^*(X))\mathbb{I}(\hat{p}_c(X) > \alpha)] + \beta_2'\mathbb{E}[(p_c^*(X) - \hat{p}_c(X))\mathbb{I}(\hat{p}_c(X) < \gamma)]$

Similar to the argument for part 1, suppose $\hat{p}$ is classwise calibrated then $\forall \alpha, \gamma$, $\mathbb{E}[(p_c^*(X) - \hat{p}_c(X))\mathbb{I}(\hat{p}_c(X) > \alpha)] = 0$ and $\mathbb{E}[(p_c^*(X) - \hat{p}_c(X))\mathbb{I}(\hat{p}_c(X) < \gamma)] = 0$; therefore it is $\mathcal{L}_{cr}$-decision calibrated.

Conversely suppose $\hat{p}$ is $\mathcal{L}_{cr}$-decision calibrated, then $\forall \alpha$ we have $\mathbb{E}[(\hat{p}_c(X) - p_c^*(X))\mathbb{I}(\hat{p}_c(X) > \alpha)] = 0$, which implies that $\hat{p}$ is classwise calibrated according to Lemma 1.

**Part 3** Choose the special loss function $\mathcal{A} = \Delta^C$ and $\ell$ as the log loss $\ell : y, a \mapsto -\log a_y$ then the Bayes action can be computed as

$$\delta_\ell(x) = \arg\inf_{a \in \Delta^C} \mathbb{E}_{\hat{Y} \sim \hat{p}(X)}[-\log a_{\hat{Y}}] = \hat{p}(x)$$

Denote $U = \hat{p}(X)$ then let $\mathcal{L}_B$ be the set of all bounded loss functions, i.e. $\mathcal{L}_B = \{\ell, |\ell(y, a)| \leq B\}$

$$\sup_{\ell' \in \mathcal{L}_B} \mathbb{E}[\ell'(Y, \delta_\ell(X))] - \mathbb{E}[\ell'(\hat{Y}, \delta_\ell(X))]$$

$$\sup_{\ell' \in \mathcal{L}_B} \mathbb{E}[\mathbb{E}[\ell'(Y, U) - \ell'(\hat{Y}, U) \mid U]]$$    Tower

$$= B\mathbb{E}[\|\mathbb{E}[p^*(X) - \hat{p}(X) \mid U]\|_1] \qquad \text{Cauchy Schwarz}$$

If $\hat{p}$ satisfies distribution calibration, then $\|\mathbb{E}[p^*(X) - \hat{p}(X) \mid U]\|_1 = 0$ almost surely, which implies that $\hat{p}$ is $\mathcal{L}_B$ decision calibrated. Conversely, if $\hat{p}$ is $\mathcal{L}_B$ decision calibrated, then $\|\mathbb{E}[p^*(X) - \hat{p}(X) \mid U]\|_1 = 0$ almost surely (because if the expectation of a non-negative random variable is zero, the random variable must be zero almost surely), which implies that $\hat{p}$ is distribution calibrated. The theorem follows because $B$ is arbitrarily chosen. $\qquad \square$

## C.2 Proofs for Section 4

**Proposition 2.** *A predictor $\hat{p}$ satisfies $(\mathcal{L}^K, \epsilon)$-decision calibration if and only if*

$$\sup_{b \in B^K} \sum_{a=1}^{K} \left\| \mathbb{E}[(\hat{Y} - Y)b(\hat{p}(X), a)] \right\|_2 \leq \epsilon \tag{8}$$

*Proof of Proposition 2.* We first introduce a new set of notations to make the proof easier to follow. Because $\mathcal{A} = [K]$ and $\mathcal{Y} \simeq [C]$, a loss function can be uniquely identified with $K$ vectors $\ell_1, \cdots, \ell_K$ where $\ell_{ac} = \ell(c, a)$. Given prediction function $\hat{p} : \mathcal{X} \to \Delta^C$ and the expected loss can be denoted as

$$\mathbb{E}_{\hat{Y} \sim \hat{p}(x)}[\ell(\hat{Y}, a)] = \langle \hat{p}(x), \ell_a \rangle \tag{11}$$

Choose any Bayes decision function $\delta_{\ell'}$ for some loss $\ell' \in \mathcal{L}^K$, as a notation shorthand denote $\delta_{\ell'}(\hat{p}(x)) = \delta_{\ell'}(x)$. We can compute the gap between the left hand side and right hand side of Definition 2 as

$$\sup_{\ell} \frac{\left| \mathbb{E}_X \mathbb{E}_{\hat{Y} \sim \hat{p}(X)}[\ell(\hat{Y}, \delta_{\ell'}(X))] - \mathbb{E}_X \mathbb{E}_{Y \sim p^*(X)}[\ell(Y, \delta_{\ell'}(X))] \right|}{\sup_a \|\ell(\cdot, a)\|_2}$$

$$= \sup_{\ell, \|\ell(\cdot, a)\|_2 \leq 1} \left| \mathbb{E}_X \mathbb{E}_{\hat{Y} \sim \hat{p}(X)}[\ell(\hat{Y}, \delta_{\ell'}(X))] - \mathbb{E}_X \mathbb{E}_{Y \sim p^*(X)}[\ell(Y, \delta_{\ell'}(X))] \right| \qquad \text{Normalize}$$

$$= \sup_{\ell, \|\ell(\cdot, a)\|_2 \leq 1} \left| \mathbb{E}_X \left[ \langle \ell_{\delta_{\ell'}(X)}, \hat{p}(X) \rangle \right] - \mathbb{E}_X \left[ \langle \ell_{\delta_{\ell'}(X)}, p^*(X) \rangle \right] \right| \qquad \text{Eq.11}$$

$$= \sup_{\ell, \|\ell(\cdot, a)\|_2 \leq 1} \left| \sum_a \mathbb{E}_X [\langle \hat{p}(X), \ell_a \rangle \mathbb{I}(\delta_{\ell'}(X) = a)] - \sum_a \mathbb{E}_X [\langle p^*(X), \ell_a \rangle \mathbb{I}(\delta_{\ell'}(X) = a)] \right| \quad \text{Total Probability}$$

$$= \sup_{\ell, \|\ell(\cdot, a)\|_2 \leq 1} \left| \sum_a \langle \mathbb{E}_X[(p^*(X) - \hat{p}(X))\mathbb{I}(\delta_{\ell'}(X) = a)], \ell_a \rangle \right| \qquad \text{Linearity}$$

$$= \sum_a \|\mathbb{E}_X[(p^*(X) - \hat{p}(X))\mathbb{I}(\delta_{\ell'}(X) = a)]\|_2 \qquad \text{Cauchy Schwarz}$$

Finally we complete the proof by observing that the set of maps

$$\{q, a \mapsto \mathbb{I}(\delta_\ell(q) = a), \ell \in \mathcal{L}^K, q \in \Delta^C\}$$

is the same as the set of maps $B^K$. We do this by establishing a correspondence where $\ell_a = -w_a/\|w_a\|_2$ then

$$\mathbb{I}(\delta_\ell(q) = a) = \mathbb{I}(\arg\inf_{a'} \langle \ell_{a'}, q \rangle = a) = \mathbb{I}(\arg\sup_{a'} \langle w_{a'}, q \rangle = a) = b_w(q, a)$$

$\qquad \square$

## C.3 Formal Statements and Proofs for Theorem 2

**Formal Statement of Theorem 2, part I.** We first define a new notation. Given a set of samples $\mathcal{D} = \{(X_1, Y_1), \cdots, (X_N, Y_N)\}$, and for any function $\psi : \mathcal{X} \times \mathcal{Y} \to \mathbb{R}$ denote $\hat{\mathbb{E}}_{\mathcal{D}}[\psi(X, Y)]$ as the empirical expectation, i.e.

$$\hat{\mathbb{E}}_{\mathcal{D}}[\psi(X, Y)] := \frac{1}{N} \sum_n \psi(X_n, Y_n)$$

**Theorem 2.1** (Formal)*. Let $B^K$ be as defined by Eq.(6), for any true distribution over $X, Y$ and any $\hat{p}$, given a set of $N$ samples $\mathcal{D} = \{(X_1, Y_1), \cdots, (X_N, Y_N)\}$, with probability $1 - \delta$ over random draws of $\mathcal{D}$,*

$$\sup_{b \in B^K} \sum_{a=1}^{K} \|\mathbb{E}[(\hat{p}(X) - Y)b(\hat{p}(X), a)]\|_2 - \sum_{a=1}^{K} \left\|\hat{\mathbb{E}}_{\mathcal{D}}[(\hat{p}(X) - Y)b(\hat{p}(X), a)]\right\|_2 \leq \tilde{O}\left(\frac{K^{3/2}C}{\sqrt{N}}\right)$$

(12)

*where $\tilde{O}$ denotes equal up to constant and logarithmic terms.*

Note that in the theorem $\delta$ does not appear on the RHS of Eq.(12). This is because the bound depends logarithmically on $\delta$.

*Proof of Theorem 2.1.* Before proving this theorem we first need a few uniform convergence Lemmas which we will prove in Appendix C.4.

**Lemma 2.** *Let $B$ by any set of functions $\{b : \Delta^C \to [0, 1]\}$ and $U, V$ be any pair of random variables where $U$ takes values in $[-1, 1]^C$ and $V$ takes values in $\Delta^C$. Let $\mathcal{D} = \{(U_1, V_1), \cdots, (U_N, V_N)\}$ be an i.i.d. draw of $N$ samples from $U, V$, define the Radamacher complexity of $B$ by*

$$\mathcal{R}_N(B) := \mathbb{E}_{\mathcal{D}, \sigma_i \sim \mathrm{Uniform}(\{-1, 1\})} \left[\sup_{b \in B} \frac{1}{N} \sum_{n=1}^{N} \sigma_i b(V_i)\right]$$

*then for any $\delta > 0$, with probability $1 - \delta$ (under random draws of $\mathcal{D}$), $\forall b \in B$*

$$\|\hat{\mathbb{E}}_{\mathcal{D}}[Ub(V)] - \mathbb{E}[Ub(V)]\|_2 \leq \sqrt{C}\mathcal{R}_N(B) + \sqrt{\frac{2C}{N}\log\frac{2C}{\delta}}$$

**Lemma 3.** *Define the function family*

$$B_a^K = \left\{b : z \mapsto \mathbb{I}(a = \arg\sup_{a'}\langle z, w_a\rangle), w_a \in \mathbb{R}^C, a = 1, \cdots, K, z \in \Delta^C\right\}$$

$$\bar{B}_a^K = \left\{b : z \mapsto \frac{e^{\langle z, w_a\rangle}}{\sum_{a'} e^{\langle z, w_{a'}\rangle}}, w_a \in \mathbb{R}^C, a = 1, \cdots, K, z \in \Delta^C\right\}$$

*then $\mathcal{R}_N(B_a^K) = O\left(\sqrt{\frac{CK\log K\log N}{N}}\right)$ and $\mathcal{R}_N(\bar{B}_a^K) = O\left(\left(\frac{K}{N}\right)^{1/4}\log\frac{N}{K}\right)$.*

Proof of the theorem is straight-forward given the above Lemmas. As a notation shorthand denote $U = \hat{p}(X) - Y$. Note that $U$ is a random vector raking values in $[-1, 1]^C$. We can rewrite the left hand side of Eq.(12) as

$$\sup_{b \in B^K} \sum_a \|\mathbb{E}[Ub(\hat{p}(X), a)]\|_2 - \|\hat{\mathbb{E}}_{\mathcal{D}}[Ub(\hat{p}(X), a)]\|_2 \tag{13}$$

$$\leq \sum_a \sup_{b \in B_a^K} \|\mathbb{E}[Ub(\hat{p}(X), a)]\|_2 - \|\hat{\mathbb{E}}_{\mathcal{D}}[Ub(\hat{p}(X), a)]\|_2 \qquad \text{Jensen} \tag{14}$$

$$\leq \sum_a \sup_{b \in B_a^K} \|\mathbb{E}[Ub(\hat{p}(X), a)] - \hat{\mathbb{E}}_{\mathcal{D}}[Ub(\hat{p}(X), a)]\|_2 \qquad \text{Triangle} \tag{15}$$

$$\leq \sum_a \sqrt{C}\mathcal{R}_N(B_a^K) + \sqrt{\frac{2C}{N}\log\frac{2C}{\delta}} \quad \text{(w.p. } 1 - \delta) \qquad \text{Lemma 2} \tag{16}$$

$$\leq \sum_a \sqrt{C}O\left(\sqrt{\frac{CK\log K\log N}{N}}\right) + \sqrt{\frac{2C}{N}\log\frac{2C}{\delta}} \qquad \text{Lemma 3} \tag{17}$$

$$\leq K\sqrt{C}O\left(\sqrt{\frac{CK\log K\log N}{N}}\right) + K\sqrt{\frac{2C}{N}\log\frac{2C}{\delta}} \tag{18}$$

$$= \tilde{O}\left(\frac{K^{3/2}C}{\sqrt{N}}\right) \tag{19}$$

$\square$

**Formal Statement Theorem 2, Part II**

**Theorem 2.2.** *Given any input $\hat{p}$ and tolerance $\epsilon$, Algorithm 1 terminates in $O(K/\epsilon^2)$ iterations. For any $\lambda > 0$, given $O(poly(K, C, \log(1/\delta), \lambda))$ samples, with $1 - \delta$ probability Algorithm 1 outputs a $(\mathcal{L}^K, \epsilon + \lambda)$-decision calibrated prediction function $\hat{p}'$ that satisfies $\mathbb{E}[\|\hat{p}'(X) - Y\|_2^2] \leq \mathbb{E}[\|\hat{p}(X) - Y\|_2^2] + \lambda$.*

*Proof of Theorem 2.2.* We adapt the proof strategy in (Hébert-Johnson et al., 2018). The key idea is to show that a potential function must decrease after each iteration of the algorithm. We choose the potential function as $\hat{\mathbb{E}}[(Y - \hat{p}(X))^2]$. Similar to Appendix A at each iteration $t$ we first lighten our notation with a shorthand $b_a(X) = b(\hat{p}^{(t-1)}(X), a)$ (at different iteration $t$, $b_a$ denotes different functions), and $b(X)$ is the vector of $(b_1(X), \cdots, b_K(X))$. If the algorithm did not terminate that implies that $b$ satisfies

$$\sum_a \|\hat{\mathbb{E}}[(\hat{p}(X) - Y)b_a(X)]\| \geq \epsilon \tag{20}$$

Denote $\gamma \in \mathbb{R}^{K \times K}$ where $\gamma_a = \hat{\mathbb{E}}[(Y - \hat{p}(X))b(X, a)]/\hat{\mathbb{E}}[b(X, a)]$. The adjustment we make is $\hat{p}'(X) = \pi(\hat{p}(X) + \sum_a \gamma_a b(X, a))$

$$\hat{\mathbb{E}}[\|Y - \hat{p}(X)\|^2] - \hat{\mathbb{E}}[\|Y - \hat{p}'(X)\|^2]$$

$$= \hat{\mathbb{E}}[\|Y - \hat{p}(X)\|^2 - \|Y - \pi(\hat{p}(X) + \sum_a \gamma_a b(X, a))\|^2]$$

$$\geq \hat{\mathbb{E}}[\|Y - \hat{p}(X)\|^2 - \|Y - \hat{p}(X) - \sum_a \gamma_a b(X, a)\|^2] \qquad \text{Projection ineq}$$

$$= \hat{\mathbb{E}}\left[(2Y - 2\hat{p}(X) - \sum_a \gamma_a b(X, a))^T \left(\sum_a \gamma_a b(X, a)\right)\right] \qquad a^2 - b^2 = (a+b)(a-b)$$

$$= 2\sum_a \gamma_a^T \gamma_a \hat{\mathbb{E}}[b(X, a)] - \sum_{a,a'} \gamma_a^T \gamma_{a'} \hat{\mathbb{E}}[b(X, a)b(X, a')] \qquad \text{Definition}\gamma_a$$

$$= 2\sum_a \gamma_a^T \gamma_a \hat{\mathbb{E}}[b(X, a)] - \sum_a \gamma_a^T \gamma_a \hat{\mathbb{E}}[b(X, a)b(X, a)] \qquad b(x, a)b(x, a') = 0, \forall a \neq a'$$

$$= 2\sum_a \gamma_a^T \gamma_a \hat{\mathbb{E}}[b(X, a)] - \sum_a \gamma_a^T \gamma_a \hat{\mathbb{E}}[b(X, a)] \qquad b(X, a)^2 = b(X, a)$$

$$= \sum_a \|\gamma_a\|^2 \hat{\mathbb{E}}[b(X, a)] \geq \frac{1}{K}\left(\sum_a \|\gamma_a\| \hat{\mathbb{E}}[b(X, a)]\right)^2 \qquad \text{Norm inequality}$$

$$= \frac{1}{K}\left(\sum_a \left\|\hat{\mathbb{E}}[(Y - \hat{p}(X)b(X, a)]\right\|\right)^2 \geq \frac{\epsilon^2}{K} \qquad \text{Definition } \gamma_a$$

Because initially for the original predictor $\hat{p}$ we must have

$$\hat{\mathbb{E}}[\|p^*(X) - \hat{p}(X)\|_2^2] \leq \hat{\mathbb{E}}[\|p^*(X) - \hat{p}(X)\|_1^2] \leq 1$$

the algorithm must converge in $K/\epsilon^2$ iterations and output a predictor $\hat{p}'$ where

$$\sum_a \|\hat{\mathbb{E}}[(\hat{p}'(X) - Y)b_a(X)]\| \leq \epsilon$$

In addition we know that

$$\hat{\mathbb{E}}[\|\hat{p}'(X) - Y\|_2^2] \leq \hat{\mathbb{E}}[\|\hat{p}(X) - Y\|_2^2]$$

Now that we have proven the theorem for empirical averages (i.e. all expectations are $\hat{\mathbb{E}}$), we can convert this proof to use true expectations (i.e. all expectations are $\mathbb{E}$) by observing that all expectations involved in the proof satisfy $\mathbb{E}[\cdot] \in \hat{\mathbb{E}}[\cdot] \pm \kappa$ for any $\kappa > 0$ and sample size that is polynomial in $\kappa$. $\square$

*Proof of Theorem 2.2'.* Observe that the matrix $D$ defined in Algorithm 2 is a symmetric, positive semi-definite and non-negative matrix such that $\sum_{a,a'} D_{aa'} = 1$. To show that the algorithm converges we first need two Lemmas on the properties of such matrices. For a positive semi-definite (PSD) symmetric matrix, let $\lambda_1$ denote the largest eigenvalue, and $\lambda_n$ denote the smallest eigenvalue (which are always real numbers). The first Lemma is a simple consequence of the Perron-Frobenius theorem,

**Lemma 4.** *Let $D$ be any symmetric PSD non-negative matrix such that $\sum_{a,a'} D_{aa'} = 1$, then $\lambda_1(D) \leq 1$, so $\lambda_n(D^{-1}) \geq 1$.*

**Lemma 5** ((Fang et al., 1994) Theorem 1)**.** *Let $D$ be a symmtric PSD matrix, then for any matrix $B$ (that has the appropriate shape to multiply with $D$)*

$$\lambda_n(D)\mathrm{trace}(B) \leq \mathrm{trace}(BD) \leq \lambda_1(D)\mathrm{trace}(B)$$

Now we can proveed to prove our main result. We have to show that the L2 error $\hat{\mathbb{E}}[(Y - \hat{p}^{(t-1)}(X))^2]$ must decrease at iteration $t$ if we still have

$$\epsilon^2/K \leq \sum_a \|\hat{\mathbb{E}}[(Y - \hat{p}^{(t-1)}(X))b_a^{(t)}(X)]\|^2 := \mathrm{trace}(RR^T)$$

We can compute the reduction in L2 error after the adjustment

$$\hat{\mathbb{E}}[(Y - \hat{p}^{(t-1)}(X))^2] - \hat{\mathbb{E}}[(Y - \hat{p}^{(t)}(X))^2]$$

$$= \hat{\mathbb{E}}\left[(2(Y - \hat{p}^{(t-1)}(X)) - R^T D^{-1} b^{(t)}(X))^T R^T D^{-1} b^{(t)}(X)\right] \qquad \text{Definition}$$

$$= 2\hat{\mathbb{E}}\left[(Y - \hat{p}^{(t-1)}(X))^T R^T D^{-1} b^{(t)}(X)\right] - \hat{\mathbb{E}}\left[b^{(t)}(X)^T D^{-T} RR^T D^{-1} b^{(t)}(X)\right]$$

$$= 2\mathrm{trace}\left(\hat{\mathbb{E}}\left[b^{(t)}(X)(Y - \hat{p}^{(t-1)}(X))^T R^T D^{-1}\right]\right)$$

$$\quad - \mathrm{trace}\left(\hat{\mathbb{E}}\left[b^{(t)}(X)b^{(t)}(X)^T D^{-T} RR^T D^{-1}\right]\right) \qquad \text{Cyclic property}$$

$$= 2\mathrm{trace}\left(RR^T D^{-1}\right) - \mathrm{trace}(RR^T D^{-1}) = \mathrm{trace}(RR^T D^{-1}) \qquad \text{Definition}$$

$$\geq \mathrm{trace}(RR^T) \geq \epsilon^2/K \qquad \text{Lemma 5 and 4}$$

Therefore, the algorithm cannot run for more than $O(K/\epsilon^2)$ iterations. Suppose the algorithm terminates we must have

$$\sup_{b \in B^K} \sum_a \|\hat{\mathbb{E}}[(Y - \hat{p}^{(t-1)}(X))b_a^{(t)}(X)]\| \leq \sup_{b \in \bar{B}^K} \sum_a \|\hat{\mathbb{E}}[(Y - \hat{p}^{(t-1)}(X))b_a^{(t)}(X)]\|$$

$$\leq \sup_{b \in \bar{B}^K} \sqrt{K}\sqrt{\sum_a \|\hat{\mathbb{E}}[(Y - \hat{p}^{(t-1)}(X))b_a^{(t)}(X)]\|^2}$$

$$\leq \sqrt{K}\epsilon/\sqrt{K} = \epsilon$$

So by Proposition 3 we can conclude that the algorithm must output a $(\mathcal{L}^K, \epsilon)$-decision calibrated prediction function. $\qquad \square$

## C.4 Proof of Remaining Lemmas

*Proof of Lemma 1.* By the orthogonal property of the condition expectation, for any event $A$ in the sigma algebra induced by $V$, we have

$$\mathbb{E}[(U - \mathbb{E}[U \mid V])\mathbb{I}_A] = 0$$

This includes the event $V > c$

$$\mathbb{E}[(U - \mathbb{E}[U \mid V])\mathbb{I}(V > c)] = 0$$

In other words,

$$\mathbb{E}[U\mathbb{I}(V > c)] = \mathbb{E}[\mathbb{E}[U \mid V]\mathbb{I}(V > c)]$$

$\qquad \square$

*Proof of Lemma 2.* First observe that by the norm inequality $\|z\|_\alpha \le C^{1/\alpha}\|z\|_\infty$ we have

$$\|\hat{\mathbb{E}}_{\mathcal{D}}[Ub(V)] - \mathbb{E}[Ub(V)]\|_2 \le \sqrt{C}\|\hat{\mathbb{E}}[Ub(V)] - \mathbb{E}[Ub(V)]\|_\infty \tag{21}$$

Denote the $c$-th dimension of $U$ by $U^c$; we now provide bounds for $|\hat{\mathbb{E}}[U^c b(V)] - \mathbb{E}[U^c b(V)]|$ by standard Radamacher complexity arguments. Define a set of ghost samples $\bar{\mathcal{D}} = \{(\bar{U}_1, \bar{V}_1), \cdots (\bar{U}_N, \bar{V}_N)\}$ and Radamacher variables $\sigma_n \in \{-1, 1\}$

$$\mathbb{E}_{\mathcal{D}}\left[\sup_b |\hat{\mathbb{E}}_{\mathcal{D}}[U^c b(V)] - \mathbb{E}[U^c b(V)]|\right] \tag{22}$$

$$= \mathbb{E}_{\mathcal{D}}\left[\sup_b \left|\hat{\mathbb{E}}_{\mathcal{D}}[U^c b(V)] - \mathbb{E}_{\bar{\mathcal{D}}}\left[\hat{\mathbb{E}}_{\bar{\mathcal{D}}}[U^c b(V)]\right]\right|\right] \qquad \text{Tower} \tag{23}$$

$$= \mathbb{E}_{\mathcal{D}}\left[\sup_b \left|\mathbb{E}_{\bar{\mathcal{D}}}[\hat{\mathbb{E}}_{\mathcal{D}}[U^c b(V)] - \hat{\mathbb{E}}_{\bar{\mathcal{D}}}[U^c b(V)]]\right|\right] \qquad \text{Linearity} \tag{24}$$

$$\le \mathbb{E}_{\mathcal{D}}\left[\sup_b \mathbb{E}_{\bar{\mathcal{D}}}\left[\left|\hat{\mathbb{E}}_{\mathcal{D}}[U^c b(V)] - \hat{\mathbb{E}}_{\bar{\mathcal{D}}}[U^c b(V)]\right|\right]\right] \qquad \text{Jensen} \tag{25}$$

$$\le \mathbb{E}_{\mathcal{D}, \bar{\mathcal{D}}}\left[\sup_b \left|\hat{\mathbb{E}}_{\mathcal{D}}[U^c b(V)] - \hat{\mathbb{E}}_{\bar{\mathcal{D}}}[U^c b(V)]\right|\right] \qquad \text{Jensen} \tag{26}$$

$$\le \mathbb{E}_{\sigma, \mathcal{D}, \bar{\mathcal{D}}}\left[\sup_b \left|\frac{1}{N}\sum_i \sigma_i U_i^c b(V_i) - \frac{1}{N}\sum_i \sigma_i \bar{U}_i^c b(\bar{V}_i)\right|\right] \qquad \text{Radamacher} \tag{27}$$

$$\le \mathbb{E}_{\sigma, \mathcal{D}, \bar{\mathcal{D}}}\left[\sup_b \left|\frac{1}{N}\sum_i \sigma_i U_i^c b(V_i)\right| + \sup_b \left|\frac{1}{N}\sum_i \sigma_i \bar{U}_i^c b(\bar{V}_i)\right|\right] \qquad \text{Jensen} \tag{28}$$

$$= 2\mathbb{E}_{\sigma, \mathcal{D}}\left[\sup_b \left|\frac{1}{N}\sum_i \sigma_i U_i^c b(V_i)\right|\right] \tag{29}$$

Suppose we know the Radamacher complexity of the function family $b$

$$\mathcal{R}_N(B) := \mathbb{E}\left[\sup_b \frac{1}{N}\sum_i \sigma_i b(V_i)\right] \tag{30}$$

Then by the contraction inequality, and observe that $U_i^c \in [-1, 1]$ so multiplication by $U_i^c$ is a 1-Lipschitz map, we can conclude for any $c \in [C]$

$$\mathcal{R}_N(B) \ge \mathbb{E}\left[\sup_b \frac{1}{N}\sum_i \sigma_i U_{ic} b(V_i)\right] \tag{31}$$

Finally observe that the map $\mathcal{D} \to \frac{1}{N}\sum_i \sigma_i U_{ic} b(V_i)$ has $2/N$ bounded difference, so by Mcdiamid inequality for any $\epsilon > 0$

$$\Pr\left[\sup_b \left|\frac{1}{N}\sum_i \sigma_i U_i^c b(V_i)\right| \ge \mathcal{R}_N(B) + \epsilon\right] \le 2e^{-N\epsilon^2/2} \tag{32}$$

By union bound we have

$$\Pr\left[\max_c \sup_b \left|\frac{1}{N}\sum_i \sigma_i U_i^c b(V_i)\right| \ge \mathcal{R}_N(B) + \epsilon\right] \le 2Ce^{-N\epsilon^2/2} \tag{33}$$

We can combine this with Eq.(21) to conclude

$$\Pr\left[\sup_b \|\hat{\mathbb{E}}[Ub(V)] - \mathbb{E}[Ub(V)]\|_2 \ge \sqrt{C}\mathcal{R}_N(B) + \sqrt{C}\epsilon\right]$$

$$\le \Pr\left[\max_c \sup_b \hat{\mathbb{E}}[U^c b(V)] - \mathbb{E}[U^c b(V)]\|_2 \ge \mathcal{R}_N(B) + \epsilon\right] \le 2Ce^{-N\epsilon^2/2}$$

Rearranging we get $\forall \delta > 0$

$$\Pr\left[\sup_b \|\hat{\mathbb{E}}[Ub(V)] - \mathbb{E}[Ub(V)]\|_2 \geq \sqrt{C}\mathcal{R}_N(B) + \sqrt{\frac{2C}{N}\log\frac{2C}{\delta}}\right] \leq \delta$$

$\square$

*Proof of Lemma 3.* For $B_a^K$ we use the VC dimension approach. Because $\forall b \in B_a^K$ the set $\{z \in \Delta^C, b(z) = 1\}$ is the intersection of $K$ many $C$-dimensional half planes, its VC dimension $\mathrm{VC}(B_a^K) \leq (C+1)2K\log_2(3K)$ (Mohri et al., 2018) (Q3.23). By Sauer's Lemma we have

$$\mathcal{R}_N(B_a^K) \leq \sqrt{\frac{2\mathrm{VC}(B_a^K)\log(eN/\mathrm{VC}(B_a^K))}{N}} = O\left(\sqrt{\frac{2CK\log K\log N}{N}}\right)$$

$\square$