# OpenReview forum: "Calibrating Predictions to Decisions: A Novel Approach to Multi-Class Calibration"
_NeurIPS.cc/2021/Conference — NeurIPS 2021 Poster_

### Official Review · Reviewer_TRQr · 2021-07-14

**Rating:** 8
**Confidence:** 4

**Summary:**

The paper defines a new notion of decision calibration and demonstrates that approximate $\mathcal{L}^K$ decision calibration is verifiable and achievable in practice, providing a practical algorithm for it. This is a result with practical calibration guarantees in multi-class calibration, which to my knowledge is the first achievement of this kind. The proposed method outperforms the existing calibration methods of temperature scaling and Dirichlet calibration in experiments on HAM10000 and ImageNet datasets and multiple architectures.

**Limitations And Societal Impact:**

The limitations and social impact have been covered adequately.


**Main Review:**

The paper is very well written and with very high significance for the field of classifier calibration. The paper demonstrates that taking the perspective of all decision-makers is a very useful one. I have the following comments, suggestions and questions.

Major:

No evidence provided for the claim 'We observe that all recalibration methods (including ours) work better if we first apply temperature scaling, hence we first apply it in all experiments.' (p.8,l.290). I would like to see results with and without temperature scaling. Given that the theoretical guarantees apply for the proposed method both with and without temperature scaling, how come there is a significant difference? Is it that the loss is comparable on validation (recalibration) data but different on test data?

It is not clear from the paper whether the experiments have been performed with a single train/validation/test split or there are more replicates with different random seeds. If there is a single split, then how are the error bars obtained in Figure 2 and lines 297-298 (p.8)?

The checklist answers '[Yes] See Appendix' for the question 'Did you include the total amount of compute and the type of resources used', but I did not find any information on this in the Appendix.

The main intuition behind Algorithm 1 has not been explained in sufficient detail: find the worst-case decision task, group by train-optimal action, shift the predictions within each group to be calibrated for that group on average, iterate. Currently it involves a lot of mathematical work to read it out from the notation.

Minor:

I suggest that the authors could offer a name for the notion of $\mathcal{L}^K$ decision calibration that would not require mathematical fonts, to be used in future research.

Although perhaps obvious, the paper does not discuss the relationship of $\mathcal{L}^K$ decision calibration with the existing notions of confidence calibration and classwise ca
libration.

Some earlier works have used cost matrices as a way to define multi-class loss functions for finite action sets. I suggest to at least mention the term 'cost matrix' because it might provide useful insights for some readers.

p.3, l.92: notation of $\Delta$ with an extra line on the left side (as if it were from mathbb font) is quite a rare choice of symbol, is this sufficiently justified?

p.4, Table 1: the associated loss functions are not unique, are they? In the sense that for each notion of calibration there exist infinitely many associated loss functions, e.g. with a different choice of a multiplicative factor but other transformations exist? I suggest to consider mentioning non-uniqueness somewhere in the paper. Does there exist a loss function associated to confidence calibration but not involving the action "I don't know"?

p.5, l.176: 'for a trivial predictors' -> 'for trivial predictors'

p.6, l.216: The L2 error is actually the Brier score, isn't it? I suggest to mention this. Could one use some other divergence measures in place of the L2 error here?

Throughout the paper: 'Radamacher' -> 'Rademacher'

p.8, l.284: 'approximately 15% of the data are used for validation' - why 'approximately'? Is it just rounding to the closest integer or something else that makes it approximate?

p.8, l.296: 'in about 5 steps' - why 'about', are there more than 1 replicate experiments? Or is this 'about' referring to the results in Appendix with different numbers of actions?

p.9, l.319: 'average calibration average calibration'

p.10, l.342: 'A Philip Dawid' -> A=?

p.14, l.476: 'Prosition' -> 'Proposition'

p.15,16, l.509,510: I suggest adding $\cdot$ in front of the identity function to separate it from the preceding symbol. Otherwise it might confuse some readers as the indicator function symbol looks a bit like a rectangle potentially modifying the previous symbol.

p.17, Eq.(12): one of the $\hat{p}$ should actually be $p^*$.

---

After discussion: I am glad to keep to grade 8.

**Time Spent Reviewing:**

6

---

> ### Author Response · Authors · 2021-08-09
> **Addressing Several Questions about our Paper**
>
> Thank you for the detailed review and suggestions!
>
> $\color{blue}{Q}$: No evidence provided for the claim 'We observe that all recalibration methods (including ours) work better if we first apply temperature scaling, hence we first apply it in all experiments.' (p.8,l.290). I would like to see results with and without temperature scaling.
>
> We ran additional experiments without temperature scaling on Imagenet (plotted below). The results suggest that with sufficiently many iterations, our decision calibration algorithm works equally well with or without temperature scaling. However, temperature scaling drastically reduces the number of iterations required for decision calibration to converge. In other words, there are computational advantages to using temperature scaling in addition to decision calibration.
>
> Test set / recalibration set performance with/without temperature scaling https://ibb.co/0QtLDTX / https://ibb.co/3kwk7B1
>
> $\color{blue}{Q}$: It is not clear from the paper whether the experiments have been performed with a single train/validation/test split
>
> The experiments are performed with random train/val/test splits. The randomness (hence the error bar) comes from both random data splits and randomness in the training process (such as random initialization).
>
> $\color{blue}{Q}$: Did you include the total amount of compute and the type of resources used
>
> We will add the following information in the Appendix: for our most expensive experiment (Imagenet-1000), each decision calibration iteration costs about 30 seconds on a 2080Ti GPU. We run up to 150 iterations, costing a total of approximately 1.5 hours. For the smaller datasets (such as HAM10000 or Imagenet-10), each decision calibration iteration costs around 10 seconds, so 150 iterations cost a total of approximately 25 minutes.
>
> $\color{blue}{Q}$: Writing suggestions
>
> Thank you. We have edited the paper according to these suggestions and they will appear in the final version.

---

### Official Review · Reviewer_JJNJ · 2021-07-16

**Rating:** 4
**Confidence:** 2

**Summary:**

This paper operates within the context of uncertainty quantification in classification. Naive generalization of the notion of a calibrated predictor from the binary to the multiclass setting is extremely prohibitive in terms of sample complexity (with respect to number of classes for the problem at hand). The authors introduce a novel notion --- decision calibration --- and analyze it theoretically and empirically on a set of prediction tasks.

**Limitations And Societal Impact:**

Connections to related works in the field should be discussed in more detail.

**Main Review:**

The paper is hard to follow, and insufficient clarity makes it hard to evaluate its significance. Findings seem to be questionable. Writing can also be improved in several directions:
1. Equations are parts of sentences, so punctuation is needed (and is missing currently).
2. Centering, commas etc. (e.g., see equation 5).
3. Line 94 --> missing fullstop.
Unfortunately, it seems to me that the paper is not well-suited for this conference.

**Time Spent Reviewing:**

3

---

> ### Author Response · Authors · 2021-08-09
> **Improved Writing**
>
> Thank you for your review. We have improved our writing, including the punctuation mistakes that you pointed out. We plan to send our paper to a copy editor to ensure that all grammar and punctuation follow professional standards.

---

### Official Review · Reviewer_WEMe · 2021-08-03

**Rating:** 7
**Confidence:** 4

**Summary:**

The paper looks at the problem of calibration for multi-class classification, with respect to downstream decision making.  The primary contribution of the paper is theoretical, providing a framework, algorithm, and analysis for decision calibration.  The decision calibration framework is intended to get around complexity issues for distribution calibration.  The authors provide some illustrative empirical evaluations to demonstrate the theoretical findings, using use cases in healthcare and image recognition.

**Limitations And Societal Impact:**

For me, the primary limitations of the paper are that there has not been an empirical evaluation of sufficient depth to uncover any potential lack of robustness.

**Main Review:**

For me, the main strength of the paper is the theoretical framing and analysis.  I appreciate the value added to the community through the discussion of weaknesses of distribution calibration.  The reframing of decision calibration is elegant and novel, and the resulting algorithm and theoretical analysis are a strong contribution.

In my view, the paper has some serious weaknesses in the empirical evaluation, and also in the overall framing in terms of being a practical approach.

In the theoretical evaluation, the method is applied and results are reported, but there are no baselines or ablations performed.  Indeed, the authors report that temperature scaling is used because it improves results, but there is no attempt to use temperature scaling on its own as a recalibration method or an ablation showing what happens when it is not applied.  Thus, the reader cannot judge whether the benefits shown are due to the recalibration method itself, or whether temperature scaling was responsible for the benefits, or whether the two together offer a unique value.  Similar comments can be made to other calibration methods, of which many appear in the literature, and at least some of which should have been tested as part of the empirical evaluation.

The focus on per-healthcare provider loss functions is maybe obscuring the point slightly – there are absolutely going to be different per-healthcare provider distributions.  While loss functions formally are defined with respect to a distribution, colloquially ML folks often focus more on the mathematical definition (such as squared loss vs. cross entropy loss) and less on the data distribution on which it is defined (such as patients admitted to Hospital X vs all people living within 5 miles of Hospital X).  This point becomes important on line 31, because the idea of matching empirical frequencies exactly only makes sense when talking about specific distributions.  Indeed, it is possible (and all too common) to have a predictor that matches empirical frequencies for training distribution D but does not match empirical frequencies on the deployment distribution D’.

Personally, I feel that it is important to evaluate recalibration methods in the presence of distributional shift, especially when healthcare is a motivating example.  Some methods such as temperature scaling have been shown to be actively detrimental under realistic conditions of distributional shift.  I'm willing to accept that this may be out of scope for this short paper, but the paper would surely be improved if this were able to be addressed in some way.

I believe that there should be at least some discussion of what happens when instead we treat k-class multiclass calibration as k-independent binary class problems, and apply binary-class calibration methods?  Why is this insufficient?

There’s a shift in notation in 2.1 compared to the intro.  Before, we we were talking about decision vectors q.  This q notation went away in the formal setup, which is a little confusing.

If I am reading correctly, this paper only focuses on multi-class single label vs. multi-class multi-label.  This should probably be called out explicitly (as the simplex would no longer be usable for multi-class multi-label).

I believe that Skin Lesions is correct (rather than Skin Legions)

In the empirical evaluation, is it really a choice of 2 actions (treatment vs. no treatment), or rather a choice of at least 14 actions (treatment, no treatment) X (7 possible conditions)?

I’m not sure what the reader is intended to gain from Figure 1 left, since there are no details given.

Fig 1 histograms – vertical axis scaling should be equal on both charts for accurate visual comparison

For the empirical evaluation, no other methods are tested.  How do we know that these improvements would not also be seen using other simple methods?

The related work section is extremely terse.

Section 7 can just be cut to make space.  There’s no need for a conclusion section of this form in a short conference paper, unless it adds new value or insights.


======

Update after reading the author responses:

I very much appreciate the clarifications, and in particular the additional empirical results exploring the question of robustness and showing strong performance there.  I will be increasing my score accordingly.




**Time Spent Reviewing:**

2

---

> ### Author Response · Authors · 2021-08-09
> **Clarifications on Empirical Evaluation and New Experiments**
>
> Dear Review WEMe,
>
> Thank you for your detailed review and suggestions. We appreciate your positive comments on our novelty and theoretical contribution. For your concern about missing experiments, we have added all the requested new experiments with code attached. The new observations further support the main claims of our paper.
>
> $\color{blue}{Q}$: There are no baselines or ablations performed. There is no attempt to use temperature scaling on its own as a recalibration method or an ablation showing what happens when it is not applied. For the empirical evaluation, no other methods are tested.
>
> In the submission, we did compare the following methods
>
> [TS] Temperature scaling on its own
>
> [TS+DC] Temperature scaling + Dirichlet calibration (main competing baseline)
>
> [TS+Ours] Temperature scaling + decision recalibration (our method)
>
> The temperature scaling results are *implicitly presented*. Specifically, we plot (in Figures 2 and 3) the performance after 0, 1, 2, ….  steps of our decision calibration algorithm [TS+Ours]. The temperature scaling results [TS] correspond to 0 steps. In Figures 2 and 3 of our submission, we can observe that [TS+Ours] performs better than [TS] or [TS+DC] on all the benchmarks. In the final version of the manuscript, we will label this baseline explicitly to avoid confusion.
>
> In addition, we added the following ablation experiments without temperature scaling
>
> [None] No recalibration
>
> [DC] Dirichlet calibration on its own
>
> [Ours] Decision calibration on its own
>
> The overall performance ranking is [TS+Ours] $\approx$ [Ours] > [TS+DC] $\approx$ [DC] >  [TS] > [None]. The results suggest that with sufficiently many iterations, our decision calibration algorithm works equally well with or without temperature scaling. However, temperature scaling reduces the number of iterations required for decision calibration to converge. In other words, there are computational advantages to using temperature scaling in addition to decision calibration. The following are the plotted results:
>
> Test set / recalibration set performance without temperature scaling https://ibb.co/0QtLDTX / https://ibb.co/3kwk7B1
>
> The code for these experiments and original logs are available at [anonymous link](https://drive.google.com/file/d/17ZCKKjmmYMQn8FrHYByYnq23WeqPnBaP/view?usp=sharing)
>
> $\color{blue}{Q}$: There are absolutely going to be different per-healthcare provider distributions. I feel that it is important to evaluate recalibration methods in the presence of distributional shift
>
> Thank you for this suggestion. Our method is primarily designed for providing guarantees assuming no distribution shift, as are almost all calibration notions. However, we do agree that robustness to distribution shift is extremely important in practice. Therefore, we added additional experiments on the Imagenet-C benchmark [1], which is a standard benchmark for assessing performance under distribution shift by corrupting the input image with 15 types of common noise.
>
> New experiment setup: We partition the Imagenet validation data into a recalibration set and a test set. We train decision calibration on the recalibration set (with no corruption), and measure performance on the test set (with corruption). In other words, the recalibration algorithms are learned on clean data but tested on corrupted data.
>
> Results: The plots are attached below. The main observation is that all methods perform worse with corruption (as expected), but our method [TS+Ours] is the best across the board on all metrics. Specifically,
>
> 1. [TS+Ours] improves decision loss estimation: With corruptions, [TS+Ours] significantly improves both the average and worst-case loss estimation error. The performance ranking is [TS+Ours] > [TS] >  [TS+DC] > [None]. Thus, decision calibration provides an effective strategy for estimating decision loss, even under distribution shifts.
>
>      (https://ibb.co/TLNn3qr https://ibb.co/Db7Hp0h https://ibb.co/Z1y3X28 ) plot the average decision loss estimation error on the (corrupted) test set on Imagenet-50, 200, 1000 respectively. The error of [None] is not visible in the plots because it is off the charts.
>
>      (https://ibb.co/dbxLMwG https://ibb.co/X2yz79H https://ibb.co/Gs574Z9 ) plot the worst decision loss estimation error on the (corrupted) test set on Imagenet-50, 200, 1000 respectively
>
>
> 2.  [TS+Ours] improves decision loss: With corruptions, [TS+Ours] improves the decision loss, and performs better than the baselines [TS] or [TS+DC] on ~80% of the benchmarks
>
>     (https://ibb.co/7rmgjR6 https://ibb.co/8xHJbTf https://ibb.co/yRKBHjz ) plot the decision loss on the (corrupted) test set on Imagenet-50, 200,1000 respectively
>
> 3. [TS+Ours] does not harm the top-1 accuracy. In fact, the accuracy on corrupted test data generally increases compared to all of the baselines [TS], [TS+DC] or [None].
>
>     (https://ibb.co/yWrkJcD) plots the accuracy on the (corrupted) test set on Imagenet-1000
>
> The code for these experiments and original logs are available at [anonymous link](https://drive.google.com/file/d/17ZCKKjmmYMQn8FrHYByYnq23WeqPnBaP/view?usp=sharing)
>
>
> $\color{blue}{Q}$: I believe that there should be at least some discussion of what happens when instead we treat k-class multiclass calibration as k-independent binary class problems? Why is this insufficient?
>
> Thank you for this suggestion. The alternative you propose is essentially class-wise calibration. In classwise calibration, we require calibration if the k-multiclass prediction is reduced to k one-vs-all binary predictions.  In the final version, we will make this comparison explicit.
>
> In the paper, we show that in both theories and experiments, decision calibration provides much stronger performance than class-wise calibration for accurate decision loss estimation. Theoretically, in Theorem 1, classwise calibration is equivalent to decision calibration for a very restricted set of decision tasks (these are the one-vs-all refrained classification tasks). Experimentally, the method designed to achieve classwise calibration (Dirichlet calibration [DC]) does not perform as well as our method.
>
> $\color{blue}{Q}$: In the empirical evaluation, is it really a choice of 2 actions (treatment vs. no treatment)
>
> Yes, it is indeed a choice of 2 actions, because ultimately the decision-maker chooses between treatment vs. no-treatment. Of course in a real medical application, there are probably several treatments available. Our illustrative example is simplified.
>
>
> [1] Benchmarking Neural Network Robustness to Common Corruptions and Perturbations. We use the entire benchmark except "frost" because there is a known technical issue, and "glass_blur" because the corruption function is too slow to finish in time for the rebuttal.

---

> ### Author Response · Authors · 2021-09-07
> **Follow up on Rebuttal**
>
> Dear Review WEMe,
>
> Thank you again for the review. Could you kindly let us know if the rebuttal addressed your concerns? To summarize our rebuttal: in the original submission we did present some of the requested experiments (some are in the Appendix); we also added the new robustness experiments.
>
> Thanks and looking forward to your reply!

---

> > ### Comment · Reviewer_WEMe · 2021-09-10
> > **thank you for the additional clarifications and results**
> >
> > I very much appreciated the detailed response, and in particular the new results showing that the proposed method performs well in the presence of distributional shift.  These results and the additional clarifications strengthen the paper significantly in my opinion.

---

> > > ### Author Response · Authors · 2021-09-10
> > > **Thank you**
> > >
> > > Thank you for reassessing the score in light of the clarifications and new results. We have updated our paper to incorporate these clarifications and new results.

---

### Decision · Program_Chairs · 2021-09-27

**Decision:**

Accept (Poster)

**Comment:**

While there was not complete consensus on this paper, two reviewers found the work to be valuable -- especially after the author responses, that included clarifications and the additional empirical results to address key concerns raised around ablations and robustness.  Additionally, I found that the most negative review was also one that provided almost no detail or reasoning, which leads me to discount this review significantly.

I will quote reviewer TRQr's private response within discussion amongst reviewers as a useful summary of the paper's strengths:

"I think that the authors have done a very good job answering all concerns raised by all reviewers and have even run very relevant additional experiments. I am definitely supporting acceptance of this paper because it fills the conceptual gap between confidence calibration and classwise calibration on one side and distribution calibration on the other side. It is not just some arbitrary mathematical generalization, it is exactly the right one in my opinion - taking the perspective of the decision maker is the best thing one can do, because that is what calibration is usually meant to serve."

I do expect that the authors will include the additional results in the final paper, and that they will use the reviewer comments to further strengthen the paper.